# HERMES: Towards Efficient and Verifiable Mathematical Reasoning in LLMs

**Azim Ospanov** [1,2]  **Zijin Feng** [1]  **Jiacheng Sun** [1]  **Haoli Bai** [1]  **Shen Xin** [3]  **Farzan Farnia** [2]

## Abstract

Informal mathematics has been central to modern large language model (LLM) reasoning, offering flexibility and efficient construction of arguments. However, purely informal reasoning is prone to logical gaps and subtle errors that are difficult to detect and correct. In contrast, formal theorem proving provides rigorous, verifiable mathematical reasoning, where each inference step is checked by a trusted compiler, but lacks the exploratory freedom of informal problem-solving. This mismatch leaves current LLM-based math agents without a principled way to combine the strengths of both paradigms. In this work, we introduce *Hermes*, the first tool-assisted agent that explicitly interleaves informal reasoning with formally verified proofs in Lean. The framework performs intermediate formal checking to prevent reasoning drift and a memory module for proof continuity across multi-step reasoning chains, enabling both exploration and verification. We evaluate *Hermes* on four challenging mathematical reasoning benchmarks using LLMs of varying parameter scales, from small models to state-of-the-art systems. Across all settings, *Hermes* reliably improves the reasoning accuracy of base models while substantially reducing reasoning token usage and computational cost compared to reward-based approaches. On difficult datasets such as AIME and HARDMath2, *Hermes*@1 achieves up to a 40% accuracy improvement while using 80% fewer total inference FLOPs. When scaled at test time, *Hermes*@5 boosts accuracy further by 20%. The implementation and codebase are publicly available at https://github.com/aziksh-ospanov/HERMES.

## 1. Introduction

In recent years, Large Language Models (LLMs) have achieved remarkable proficiency in mathematical reasoning (Wei et al., 2022; Kojima et al., 2022; Yu et al., 2023; Xu et al., 2024), with some systems even demonstrating the potential to solve Olympiad-level problems (Huang & Yang, 2025). A key advancement driving this progress is the Chain-of-Thought (CoT) approach, which enables LLMs to plan step-by-step reasoning, decompose complex problems into sub-goals, generate intermediate reasoning steps, and iteratively assess and correct them. However, long CoTs remain susceptible to logical leaps, subtle errors, and hallucinations, stemming from incomplete domain knowledge, imprecise reasoning, or the accumulation of small mistakes over multiple steps (Huang et al., 2025; Lu et al., 2025; Akbar et al., 2024). These issues could lead to unstable reasoning and degraded performance, and when uncertain, LLMs may produce overly long or repetitive reasoning traces, increasing token usage without necessarily improving correctness.

To address these limitations, researchers have proposed Process Reward Models (PRMs) and Outcome Reward Models (ORMs)(Lightman et al., 2024; Wang et al., 2024b), which aim to guide LLM reasoning toward correctness by scoring intermediate reasoning steps or final solutions. Specifically, PRMs assign a score at each reasoning step, rewarding local correctness, while an ORMs provide a single scalar reflecting overall solution quality. While these models can improve reasoning performance, they function largely as black-box evaluators that assign numerical scores without explaining why a reasoning trajectory is valid or flawed, offering limited interpretability and no explicit verification of mathematical correctness (She et al., 2025). Moreover, their training requires substantial human curation (Huang et al., 2025), and automated supervision methods introduce noise by inferring step correctness labels from final answers (Zhang et al., 2025b), leading to misalignment with true stepwise correctness. Ultimately, because both PRMs and ORMs rely on LLMs as evaluative backbone, their reward signals inherit the stochasticity, biases, and instability of LLM-based judgment (Shao et al., 2024).

In parallel to these, another line of work has focused on formal theorem proving, which relies on proof assistants with trusted kernels such as Lean4 (Moura & Ullrich, 2021),

---

[1]Huawei Foundation Model Department [2]Department of Computer Science & Engineering, The Chinese University of Hong Kong [3]Celia Team. Correspondence to: Azim Ospanov <aospanov9@cse.cuhk.edu.hk>, Zijin Feng <fengzijinn@gmail.com>.

*Proceedings of the 43rd International Conference on Machine Learning*, Seoul, South Korea. PMLR 306, 2026. Copyright 2026 by the author(s).

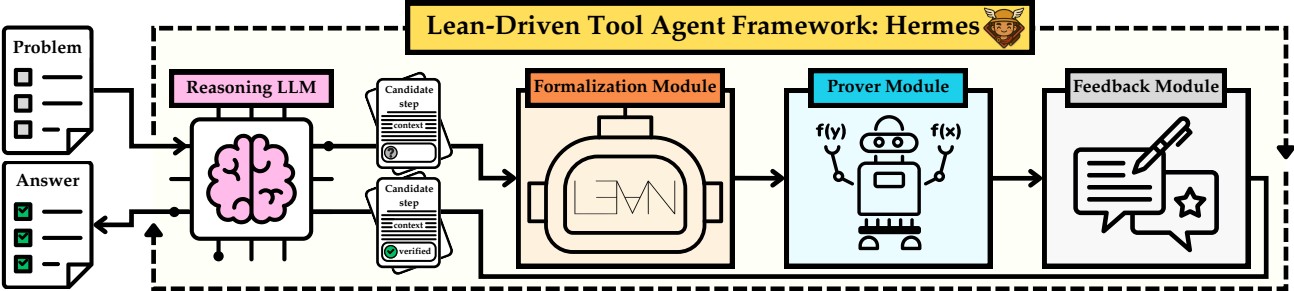

*Figure 1.* Overview of *Hermes* framework. *Hermes* is a Lean4-driven, multi-modular reasoning agent integrating LLM reasoning with formal verification for reliable mathematical problem solving. It comprises four modules: an LLM that generates reasoning steps, a formalizer that formalizes these steps into Lean code, a prover that symbolically verifies their correctness, and a feedback module that returns verification signals for subsequent reasoning. This design enables iterative reasoning with improved correctness and efficiency.

Coq (Bertot & Castéran, 2013), and Isabelle (Nipkow et al., 2002). These systems enforce rigorous formal, machine-checkable reasoning, in contrast to the informal reasoning of traditional mathematics expressed in natural language. Recently, formal language-based systems such as AlphaProof and AlphaGeometry (Yang et al., 2025; Google DeepMind, 2024; Trinh et al., 2024) have achieved remarkable success on International Mathematical Olympiad (IMO) problems, rivaling top human performers. Their principal strength lies in verifiability and immunity to hallucinations, as formal verification is embedded directly into the proof search process, ensuring that inference is rigorously justified.

Inspired by the strengths of both paradigms, in this work, we bridge the gap between formal and informal mathematics by combining the verifiability of formal reasoning with the flexibility and expressiveness of LLM-based informal reasoning. We introduce *Hermes*, a multi-modular scalable tool-augmented agent (abbreviated from "**H**ybrid Ag**E**nt for **R**easoning in **M**athematics with N**E**uro-**S**ymbolic Lean4 verification"), designed to integrate formal verification into the LLM reasoning process. *Hermes* leverages modern LLMs' tool-calling capability to verify individual reasoning steps during inference and builds a memory block that ensures continuity of proof claims. For each critical proof step, the agent formalizes the natural-language statement into Lean goal, verifies formalization consistency through back-formalization, and invokes a prover module to attempt a proof or counter-proof. The resulting formal signal is then fed back into the LLM to inform its next reasoning step. The overview of our agentic framework is illustrated in Figure 1. We show that incorporating *Hermes* significantly enhances LLM accuracy across mathematical reasoning benchmarks of varying difficulty. It reduces token usage compared to traditional score-based methods and provides interpretable, step-level correctness feedback, offering transparency into how reasoning paths evolve and why certain trajectories lead to valid conclusions while others result in hallucinations. Our contributions are as follows:

- We develop the first tool-based Lean4 reasoning agent that verifies feasible intermediate proof steps during inference, providing LLMs with kernel-based correctness signals for mathematical reasoning.

- We introduce a Lean4-powered memory block that accumulates and validates intermediate claims in context, ensuring cross-step consistency and reducing the propagation of errors in long reasoning chains.

- *Hermes* improves average accuracy by 23% across four benchmarks, notably achieving 40% higher accuracy on AIME'25 with 80% less compute. We demonstrate scalability by showing additional performance gains when integrating *Hermes* with Best-of-N sampling.

## 2. Related Works

**Automatic Theorem Proving.** Recent LLM-based provers have achieved remarkable advances in formal reasoning within Lean4. Goedel-Prover-V2 (Lin et al., 2025b) introduces large-scale, verifier-guided training with scaffolded data synthesis to achieve state-of-the-art accuracy on MiniF2F and PutnamBench. Kimina-Prover (Wang et al., 2025b;a) emphasizes structured reasoning patterns and reinforcement learning to improve sample efficiency. DeepSeek-Prover-V2 (Xin et al., 2025) focuses on subgoal decomposition, using hierarchical reasoning to bridge informal and formal proofs. Seed-Prover (Chen et al., 2025) advances lemma-style reasoning and multi-tier inference, achieving competition-level results on IMO and Putnam benchmarks. Collectively, these systems highlight a substantial body of work on automated theorem proving.

**Autoformalization.** Recent work in autoformalization, translating informal mathematical statements into formal statements, has made rapid advancements. For example, Goedel-Autoformalizer (Lin et al., 2025b) built a dataset of 1.64 million formal statements from natural-language problems in Lean4 and used this to train a high-performing autoformalizer model. Mean-

while, Mathesis-Autoformalizer (Yu et al., 2025) introduces a reinforcement-learning framework with a novel "LeanScorer" for assessment, and shows accuracy gains on the Gaokao-Formal benchmark. The Kimina-Autoformalizer (Wang et al., 2025b;a) similarly converts natural-language problems into Lean4 statements via a fine-tuned LLM and expert iteration. Finally, HERALD-Translator (Gao et al., 2025a) provides a large annotated Lean4 corpus ($\approx$580K statements) by back-translating parts of Mathlib and demonstrates high accuracy in miniF2F (Zheng et al., 2022) benchmark. According to previous research efforts, the current generation of formalization models are capable of translating informal statements to formal goals with syntactic and semantic correctness.

**LLM collaboration with external experts.** Another prominent line of research focuses on using external feedback to guide LLMs toward producing correct responses. Specifically, APOLLO (Ospanov et al., 2025) proposed a model-agnostic proof-repair framework that enhances sample efficiency and reliability without increasing model size. (Ringer, 2021) introduced a Coq compiler-based repair agent that adapts to changes in the underlying definitions. Moreover, (Jiang et al., 2022) show that combining the strengths of built-in solvers (e.g., Sledgehammer in Isabelle, built-in tactics in Lean) with LLMs leads to performance gains. (First et al., 2023; Wang et al., 2025a; Lin et al., 2025b; Yousefzadeh et al., 2025) report that incorporating compiler feedback encourages LLMs to adapt and correct previously generated proofs. This behavior is observed in general-purpose models but can be further amplified by appending compiler feedback during the fine-tuning stage for dedicated theorem proving LLMs.

**Enhanced Reasoning Techniques.** To further advance LLM reasoning, score-based models have emerged that evaluate the chain of thought produced by reasoning models. The goal is to identify and prioritize the most effective reasoning traces. Two main approaches exist: Outcome Reward Models (ORMs), which assess only the final answer, and Process Reward Models (PRMs), which score reasoning steps sequentially. (Lightman et al., 2024; Wang et al., 2024b) Furthermore, Safe (Liu et al., 2025) proposed training a small LSTM reward model based on Lean4 verified reasoning traces. Another line of work focuses on tool-based agents that equip LLMs with more complex functionality and the ability to query and verify their own outputs. Numerous studies have shown that teaching LLMs to use specific tools enhances their performance on given tasks (Qu et al., 2025; Qin et al., 2023), including reasoning. (Gao et al., 2025b) introduced Chain-of-Abstraction to better leverage tools in multi-step reasoning, while (Ning et al., 2024) examined the importance of using tools in the right scenarios to improve reasoning in LLMs.

**Neurosymbolic Solvers and LLMs.** Recent work has advanced neuro-symbolic reasoning by integrating LLMs with formal logic and theorem proving. LogicLM (Pan et al., 2023) and LogicLM++ (Kirtania et al., 2024) translate natural-language problems into symbolic form and iteratively refine them for solver-based verification. DSP (Jiang et al., 2023) and DSP+ (Cao et al., 2025) bridge informal reasoning and formal proofs through a draft–sketch–prove pipeline. Lean-STaR (Lin et al., 2025a) enhances formal reasoning by interleaving natural-language rationales with Lean tactics, while DTV (Zhou et al., 2023) explores deterministic verifier-guided reasoning. Collectively, these methods demonstrate the effectiveness of hybrid LLM–symbolic approaches for logical and mathematical reasoning.

## 3. Problem Formulation

We consider verifiable mathematical problem solving in natural language. Given a problem $p$, a language model produces a sequence of $T$ intermediate reasoning steps $S = (s_0, s_1, \cdots, s_T)$, where each step $s_t$ is a natural-language mathematical claim. Reasoning proceeds autoregressively, with each step conditioned on the previous context. A subset of steps, informally termed critical proof steps, are those whose correctness is essential for the validity of the final solution. We assume access to a formal verification mechanism that can be applied to a selected subset of such steps, denoted by $\tilde{S} \subseteq S$, with typically $|\tilde{S}| \leq T$.

The objective is to design an inference-time reasoning method that leverages formal theorem proving to improve step-level correctness and overall reasoning efficiency, while preserving the exploratory flexibility of language-based reasoning, ultimately producing a correct answer to $p$.

## 4. Our Approach

In this section, we describe *Hermes*, our framework for a verifiable and interpretable tool-based agent for mathematical reasoning. *Hermes* is designed to be split into multiple swappable modules that analyze intermediate proof steps and produce proper verification for each mathematical step. Figure 2 illustrates the complete agent pipeline, including intermediate proof transformations after each module. At the core of our agent lies the tool-calling capability of modern LLMs, which allows the model to pause generation and query external agents to verify reasoning steps or retrieve additional context. In our design, the LLM is instructed to verify every critical math step contributing to the final proof.

Our Lean backend of choice is the Lean4 REPL (Lean FRO, 2023), which provides compatibility with Python-based tool integration. We also employ the verification scheduler, which parallelizes Lean code verification for faster inference.

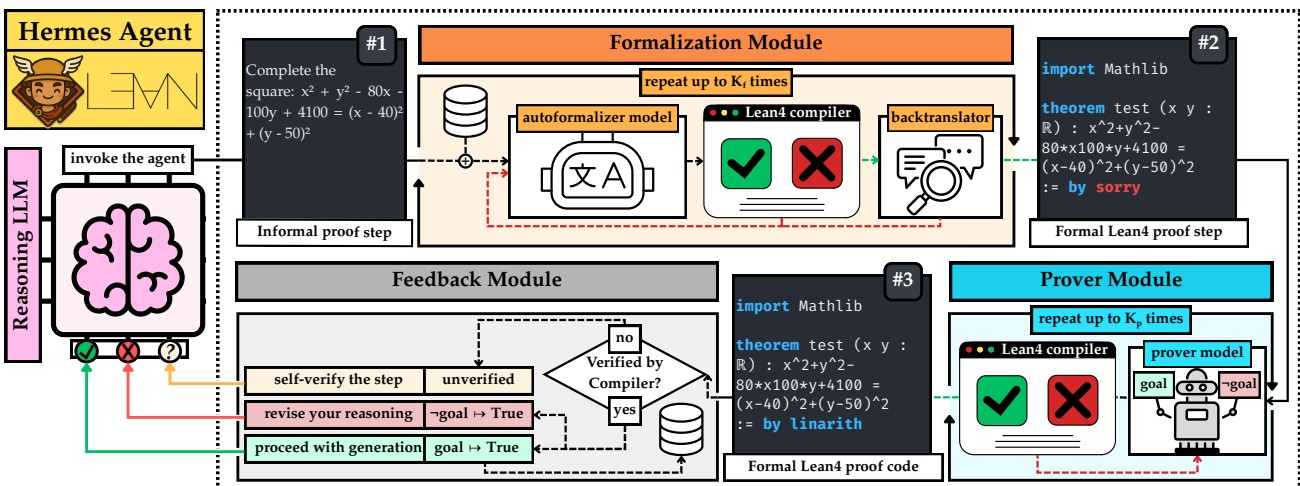

*Figure 2.* Full *Hermes* framework with illustrative examples.

The REPL validates proofs generated by the prover module, returning a simple `True/False` signal that indicates the correctness of the intermediate proof step. Additionally, our memory block uses embedding-based top-k retrieval of the most relevant contextual information. The following subsections describe each *Hermes* module in detail.

### 4.1. Formalization Module

The formalization module is responsible for accurately formalizing each proof step expressed in natural language into its equivalent Lean4 statement. At this stage, we employ a dedicated Lean autoformalizer model that generates a Lean statement corresponding to the informal proof step, inserting a `sorry` placeholder. To ensure the correctness and semantic relevance of the formalization, we apply two-step verification. First, the formalized statement must successfully compile in Lean. This verification guarantees that the prover module is provided with valid code, and well-defined premises and goals. Second, the validated Lean code is back-formalized into informal language, and the LLM evaluates the equivalence between the original and back-formalized proof steps. If the two are deemed equivalent, the formalization module forwards the Lean statement to the prover module. The number of attempts to obtain a valid formalization is controlled by the parameter $K_f$, which defines the formalization budget. A higher $K_f$ typically yields more accurate formalizations but may increase the agent's runtime. This hyperparameter is user-controlled. This module is highlighted in orange in Figure 2.

### 4.2. Prover Module

The prover module receives a valid Lean statement and attempts to find a corresponding proof within Lean. At this stage, we incorporate a whole-proof generation model together with built-in Lean solvers; however, any proof-generation method, such as tree-based provers or premise-

selection theorem provers, is compatible with our agent. This module attempts two proofs in parallel: the original goal and its negation, ¬goal. The objective is to either prove the original statement or derive a counter-proof (e.g., a counterexample) to demonstrate the invalidity of the step. Similar to the formalization module, the prover operates under a sampling budget controlled by the parameter $K_p$. We observe that increasing $K_p$ improves performance but incurs additional inference cost, as the prover must explore more proof attempts before reaching a conclusion. Once the prover finds a proof or counter-proof, it returns this verification signal to the LLM to guide the subsequent reasoning chain. This module is highlighted in blue in Figure 2.

### 4.3. Memory Block

Memory block is responsible for collecting all validated proof steps in a structured database. Each proof step is stored alongside its vector representation to enable fast, similarity-based retrieval. Because some problems involve long chains of thought (CoT) with branching, non-overlapping reasoning paths, we implement a top-$k$ retrieval mechanism to select the most relevant memories for the currently evaluated proof step. Furthermore, to maintain proof continuity and prevent the model from pursuing irrelevant or disjoint reasoning paths, the retrieved memories are passed as additional context to the formalization module. These are incorporated into the formalized Lean statement as preliminary hypotheses, ensuring logical consistency across proof steps. The memory block corresponds to the "database" icon in Figure 2.

### 4.4. Lean4 REPL Feedback

Our agent returns one of three states: (1) proof succeeded, (2) counter-proof found, or (3) verification failed. The first two states drive the next-generation strategy: the LLM either revises its reasoning when hallucination is detected or

continues exploring already-verified steps. The third state arises when the autoformalizer fails to produce a valid formalization, or the prover cannot prove/disprove the goal. The design is compatible with any autoformalizer–prover pairing, so as research progresses, stronger components can be swapped in seamlessly. This flexibility ensures the design remains relevant for non-mathematical applications and verification environments beyond Lean4. Given feedback, the LLM then chooses to (1) continue exploring the current line of reasoning, (2) attempt an alternative approach, or (3) proceed cautiously and self-verify reasoning, since the step did not pass the Lean4 compiler.

### 4.5. Complexity Analysis

In this section, we analyze the token-level inference complexity of *Hermes* and compare it with standard CoT and reward-model–based baselines (ORM and PRM). We show that *Hermes* is theoretically no more expensive than PRM and lies between ORM and PRM in complexity.

**Theorem 4.1.** *Consider a reasoning process that generates a total of $N$ reasoning tokens. Suppose verification is performed at $T$ reasoning steps, and that each verification invocation executes $K$ independent attempts. Let $n_f$, $n_p$ denote the average number of tokens generate per attempt by the formalizer and prover, respectively. Then the overall token-level inference complexity of the reasoning–verification process is $\mathcal{O}(N^2 + KTN^2)$.*

*Proof.* The total token complexity consists of a reasoning cost and a verification cost. For reasoning, under standard autoregressive decoding for a Transformer, generating the $i$-th token incurs an incremental cost $\mathcal{O}(i)$ proportional to the context length. Summing over all $N$ tokens yields a total reasoning cost of $\sum_{i=1}^{N} \mathcal{O}(i) = \mathcal{O}(N^2)$. For verification, at each of the $T$ steps, the formalizer generates candidate formalized statements and the prover attempts corresponding proofs. Each action takes $K$ formalization–proving attempts. The per-step verification cost is thus $\mathcal{O}(K(n_f^2 + n_p^2))$. Summing over $T$ steps yields a total verification cost of $\mathcal{O}(KT(n_f^2 + n_p^2)) = \mathcal{O}(KTN^2)$. Thus, the total complexity of the process is $\mathcal{O}(N^2 + KTN^2)$. □

Theorem 4.1 provides a unified characterization of token complxity across different methods. For CoT reasoning, no verification is performed and the complexity reduces to $\mathcal{O}(N^2)$. For ORM- and PRM-based methods, $K$ candidate reasoning traces are first generated before verification. ORM-based methods apply outcome-level verification once per trace ($T = 1$), leading to $\mathcal{O}(KN^2)$ complexity. PRM-based methods perform verification at every step, leading to $\mathcal{O}(TKN^2)$. In contrast, *Hermes* differs by generating a single reasoning trace and interleaving verification during inference. In practice, verification is applied at a small subset of critical steps, whose number is denoted by $\tilde{T}$ and $\tilde{T} \ll T$.

Each verification executes $K$ attempts, with $K$ is comparable to that used in ORM and PRM. Consequently, the complexity of *Hermes* is $\mathcal{O}(\tilde{T}KN^2)$, yielding the ordering

$$\mathcal{O}_{ORM}(KN^2) \leq \mathcal{O}_{Hermes}(\tilde{T}KN^2) \leq \mathcal{O}_{PRM}(TKN^2)$$

This analysis shows that *Hermes* is theoretically no more expensive than PRM. Empirically, *Hermes* achieves much higher accuracy, indicating superior overall effectiveness and token-efficiency.

## 5. Experiments

### 5.1. Experimental Setup

***Hermes* setup.** For *Hermes* evaluation, we set both $K_p$ and $K_f$ to 4, which strikes a good balance between inference speed and downstream Lean verification accuracy. Lean timeout is set to 60 seconds. The memory module selects the top three recorded steps as context. The prover of choice is Goedel-Prover-V2-8B (Lin et al., 2025b) and the autoformalizer model is Goedel-Autoformalizer-8B (Lin et al., 2025b), unless stated otherwise. Chosen embedding model for the memory block is Qwen3-Embedding-0.6B (Zhang et al., 2025a). To ensure consistency with the Goedel-Autoformalizer and Goedel-Prover training data, we utilize Lean version v4.9.0.

**Baseline methods.** We compare our approach against several strong baselines, with Safe (Liu et al., 2025) serving as the previous state-of-the-art method. We further introduce Safe*, an improved variant that replaces the prover and autoformalizer in Safe with more advanced model Goedel-Prover-V2 (Lin et al., 2025b) and Goedel-Autoformalizer (Lin et al., 2025b) to match our *Hermes* setup. In addition, we include CoT, a zero-shot chain-of-thought baseline, and Majority vote, which selects the most frequent answer from five sampled reasoning traces. We also include a range of outcome and process reward models. The outcome reward models include Skywork-Reward-Llama-3.1-8Bv0.2 (Liu et al., 2024) and ArmoRM-Llama3-8Bv0.1 (Wang et al., 2024a), while the process reward models include math-shepherd-mistral-7b-prm (Wang et al., 2024b) and RLHFlow/Llama3.1-8B-PRMDeepseek-Data (Dong et al., 2024). The experiments are conducted on numerous base models: Qwen3-8B (Team, 2025) and OpenAI o3-mini (OpenAI, 2025) in "thinking mode" and DeepSeek-V3.1 (et al., 2025) and V3.2 (DeepSeek-AI et al., 2025) in "non-thinking" mode. We selected strong models from both generation styles to assess our agent's compatibility with mainstream LLMs.

**Datasets and evaluation.** We conduct experiments on four widely used math benchmarks: MATH-500 (Lightman et al., 2023), AIME'25 (Competitions & of Problem Solving, 2025), CollegeMath (Tang et al., 2024), and HARD-

*Table 1.* Accuracy (%) of different reasoning models under four inference strategies: zero-shot CoT (@1), majority vote (@5), reward-model selection (Best-of-5), and *Hermes* (@1). Results are reported on four benchmarks: MATH500 (MATH), AIME'25 (AIME), CollegeMath (CM), and HARDMath2 (HM2). The best and second-best results are **bolded** and underlined, respectively.

| | Qwen3-8B | | | | OpenAI o3-mini | | | | DeepSeek-V3.2 | | | |
|---|---|---|---|---|---|---|---|---|---|---|---|---|
| | MATH | AIME | CM | HM2 | MATH | AIME | CM | HM2 | MATH | AIME | CM | HM2 |
| CoT@1 | 84.8 | 20.0 | 69.1 | 4.3 | 95.8 | 63.3 | 75.2 | 23.2 | 95.8 | 50.0 | 79.8 | 25.6 |
| CoT@5+Majority | 87.0 | 16.7 | 70.3 | 4.7 | 96.8 | 70.0 | 76.0 | 22.7 | 97.6 | 60.0 | 80.8 | 26.1 |
| CoT@5+Skywork | 91.0 | 30.0 | 72.0 | 5.7 | 96.8 | 83.3 | 76.1 | 29.4 | 97.4 | 53.3 | 80.4 | 30.8 |
| CoT@5+ArmoRM | 88.6 | 30.0 | 72.4 | 5.2 | 96.2 | 76.7 | 76.1 | 28.4 | 97.2 | 53.3 | 80.3 | 28.4 |
| CoT@5+Shepherd | 87.8 | 23.3 | 70.2 | 5.7 | 96.4 | 80.0 | 75.7 | 25.6 | 97.0 | 60.0 | 80.4 | 27.8 |
| CoT@5+RLHFlow | 84.0 | 20.0 | 69.4 | 5.7 | 95.8 | 70.0 | 75.9 | 28.4 | 97.2 | 60.0 | 80.3 | 24.6 |
| *Lean-based methods* | | | | | | | | | | | | |
| CoT@5+Safe | 89.4 | 23.3 | 72.4 | 5.7 | 96.0 | 83.3 | 75.7 | 26.1 | 96.2 | 56.7 | 80.0 | 28.9 |
| CoT@5+Safe* | 89.4 | 23.3 | 72.5 | 6.2 | 96.8 | 83.3 | 75.8 | 25.6 | 96.6 | 60.0 | 80.1 | 30.3 |
| *Hermes@1* | 91.2 | 30.0 | 73.0 | 6.6 | **97.2** | **86.7** | 78.9 | 31.3 | 98.4 | 70.0 | 83.3 | 33.6 |
| *Hermes@5+Majority* | 93.0 | 33.3 | 75.8 | 7.6 | 97.0 | **86.7** | **79.4** | **31.8** | **98.6** | 76.7 | **85.2** | 35.1 |
| *Hermes@5+Skywork* | **94.6** | **43.3** | **77.4** | **10.0** | 97.0 | **86.7** | 79.2 | 31.3 | 98.2 | **80.0** | 84.0 | **36.5** |

Math2 (Roggeveen et al., 2025). For brevity, we refer to them as MATH, AIME, CM, and HM2, respectively, throughout the experiments. Each problem has a 15-minute time limit and an 8,192-token budget (prompt + generation). All *Hermes* accuracy reports are @1, i.e., the reasoning models were given only one attempt per problem. For score-based methods, we evaluate each problem with $N$ generated candidates (CoTs) and report Best-of-$N$ (BoN). BoN accuracy is computed based on the reasoning trace selected by the reward model from the $N$ candidates. Unless stated otherwise, $N$ is set to 5 reasoning traces, consistent with the configuration adopted in Safe (Liu et al., 2025).

### 5.2. Performance of *Hermes* against base LLMs and reward-based generation

Table 1 reports the accuracy of eight baseline inference methods and our proposed *Hermes* across three reasoning models and four benchmark datasets, with each method evaluated on every combination of model and dataset. We selected representative reasoning models from three distinct parameter scales: a sub-10B model (Qwen3-8B), a medium-sized model (o3-mini), and a large model (DeepSeek-V3.2), each from a distinct provider. Compared with non–Lean-based methods, *Hermes* consistently outperforms all baselines across models and benchmarks, except on the AIME benchmark with Qwen3-8B, where Skywork and ArmoRM achieve comparable performance. Specifically, averaged across all reasoning models and benchmarks, *Hermes* surpasses zero-shot CoT, Majority vote, ORM-based Skywork and ArmoRM, and PRM-based Shepherd and RLHFlow methods by 23.4%, 23.9%, 5.7%, 8%, 12.2%, and 14.9%, respectively. Compared with the Lean-based baselines Safe and Safe*, *Hermes* achieves consistently higher accuracy across all benchmarks, exceeding them by an average of 10.3% and 8.5%, respectively. This demonstrates *Hermes*'s

more effective integration of neural reasoning and symbolic verification. We also observe that on more challenging datasets, *Hermes* achieves the largest relative improvements. For instance, on the harder dataset HM2, Deepseek-V3.2+*Hermes* improves performance by 31.5%, whereas on the easier MATH dataset, the gain is only 2.7% compared to CoT. This suggests that *Hermes* is particularly effective at correcting errors in settings that require more advanced mathematical reasoning. Moreover, we experimented with combining *Hermes* with a reward model. Similar to CoT@5+Skywork, we sampled 5 reasoning traces produced by *Hermes* and applied BoN selection using Skywork. The results are shown in the last row of Table 1. When combined with a reward model, the final accuracy increased by 10.8% on average compared to *Hermes*@1. Since *Hermes* produces a standard chain-of-thought, it remains compatible with previously developed reasoning trace methods.

### 5.3. Token budgets and generation efficiency

As shown in Figure 3, our evaluation indicates that the reasoning token budget of *Hermes* is comparable to that of zero-shot chain-of-thought, while being 4–6 times lower than score-based methods such as Safe, ORMs, and PRMs. These results demonstrate that incorporating intermediate, verifiable feedback not only improves accuracy across a range of benchmarks but also significantly reduces token budgets compared to BoN reasoning trace selection.

To further investigate the performance of score-based methods, we conducted an additional experiment in which we increased the number of sampled reasoning traces from 1 (zero-shot chain-of-thought) to 60. Figure 4 reports BoN accuracy across four inference methods: Safe, Safe*, math-shepherd-mistral-7b-prm, and ArmoRM-Llama3-8Bv0.1.

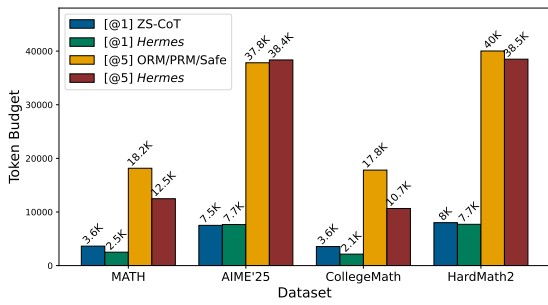

(a) Qwen3-8B

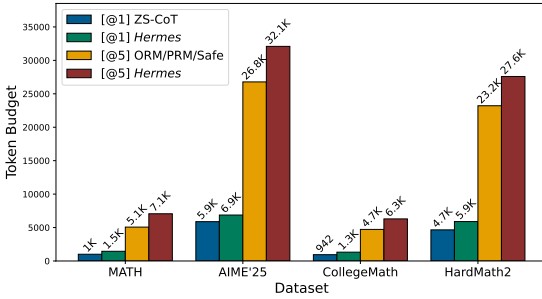

(b) OpenAI o3-mini

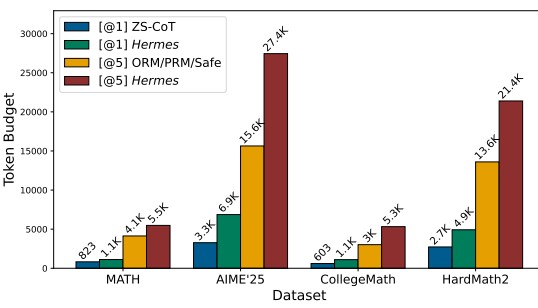

(c) DeepSeek-V3.2

*Figure 3.* Average *reasoning* token usage per problem on MATH500, AIME'25, CollegeMath, and HardMath2 under Zero-Shot Chain-of-Thought, *Hermes*, and Reward-based Best-of-5 settings.

*Hermes* is fixed at @1 and is shown as a dashed green line. Using DeepSeek-V3.1 as the base reasoning model, we observe that scaling N does not yield significant gains in accuracy. All reward-based methods underperform compared to *Hermes*, while consuming a linearly increasing number of tokens. This suggests that, rather than repeatedly regenerating reasoning traces, it is more effective to incorporate higher-quality intermediate verification and guidance mechanisms that enhance reasoning quality without incurring additional token costs. On the other hand, the same experiment on Qwen3-8B shows that reward-based models can outperform *Hermes* when given a sufficiently large token budget. We observe that for $n \geq 20$, both Safe* and ArmoRM exceed 92% accuracy; however, they consume roughly 28 times more tokens than *Hermes*. We hypothesize

that reward-based BoN is more effective when the quality of reasoning traces exhibits higher variance, as is the case for smaller models such as Qwen3-8B.

## 5.4. Computational Efficiency Analysis

Following the methodology of (Kaplan et al., 2020), we further examine the computational efficiency of *Hermes* in comparison to CoT and reward-based approaches. Because our agent requires communication among the reasoning, formalization, and prover models, we estimate the total computational cost in terms of FLOPs for two representative models: Qwen3-8B and DeepSeek-V3.1. OpenAI o3-mini is excluded from this analysis, as its architecture and parameter count are not publicly available. We approximate the FLOP budget using the formula $C_{forward} = 2N + 2n_{layer}n_{ctx}d_{attn}$, where $N$ denotes the number of model parameters, $n_{layer}$ the number of layers, $n_{ctx}$ the input context length (set to the maximum of 8,192 tokens in our experiments), and $d_{attn}$ the dimensionality of the attention output. Values for $N$, $n_{layer}$, and $d_{attn}$ were derived from each model's configuration files. As shown in Figure 5, even after accounting for the additional computation required by the *Hermes* agent, its total cost remains comparable to zero-shot CoT and substantially lower than that of reward-based models. These results demonstrate that, despite involving external formalization and theorem-proving modules, *Hermes* maintains strong computational efficiency and effective resource utilization.

## 5.5. Ablation studies on different *Hermes* modules

Recalling Section 4, *Hermes* is designed as a multi-modular agent. In this section, we conduct an ablation study to evaluate the contribution of its two key modules: the memory buffer and the prover module. The formalization and feedback modules are not ablated, as they serve as essential intermediaries bridging informal and formal mathematical reasoning. When ablating the memory buffer, *Hermes* no longer stores intermediate proof steps, preventing both the formalization module and prover module from accessing previously validated information necessary for maintaining coherent reasoning across steps. When ablating the prover module, the agent omits a query to the dedicated prover model and instead relies solely on the built-in Lean solvers, similar to the AutoSolver module in (Ospanov et al., 2025).

Table 2 presents ablation results on MATH and AIME using DeepSeek-V3.1. Removing the prover causes the largest performance drop (e.g., 66.7% → 50.0% on AIME) due to less capable proof search engine, while emoving the memory buffer disrupts reasoning continuity. Notably, the harder AIME benchmark is far more sensitive to these ablations than MATH (25.0% vs. 4.5% relative drop), as its longer reasoning chains amplify error propagation. These results indicate that the advantages of *Hermes*'s modular design be-

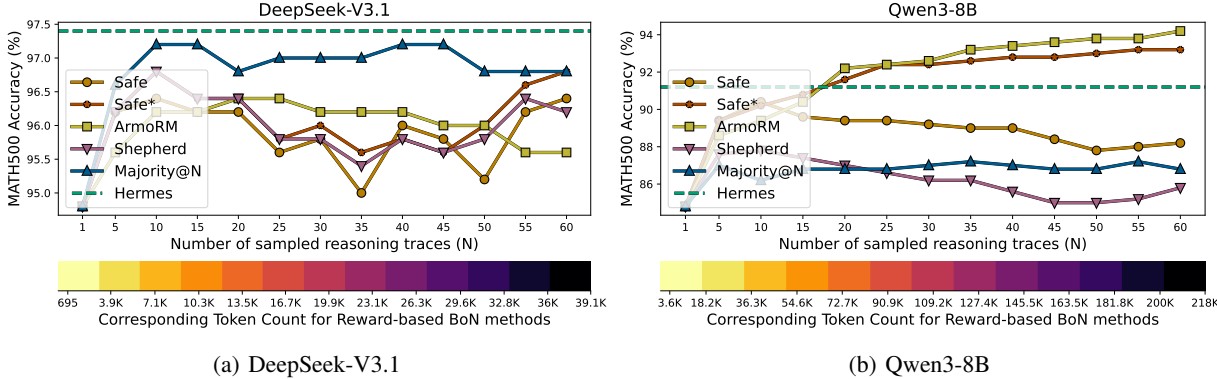

*Figure 4.* Scaling behavior of BoN across ORM, PRM, Safe and Majority vote. The green dashed line indicates *Hermes* performance at @1. The corresponding token consumption for each BoN is shown below as a one-dimensional heatmap.

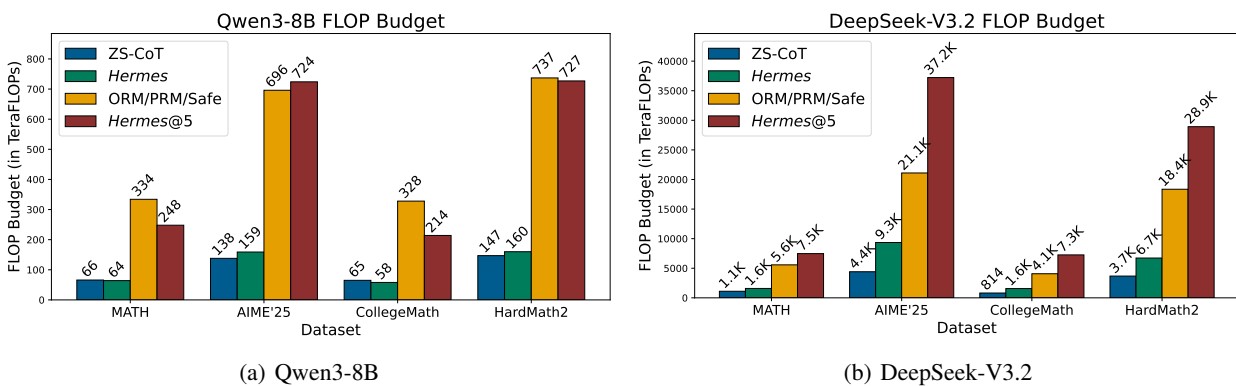

*Figure 5.* Average TeraFLOPs per problem on MATH500, AIME'25, CollegeMath, and HardMath2 under Zero-Shot Chain-of-Thought, *Hermes*@1, Reward-Based Best-of-5 and *Hermes*@5 settings. Energy usage was recorded for each of the participating models.

*Table 2.* Ablation study of different sub-modules of *Hermes* . ✓ indicates inclusion of a module; ✗ indicates exclusion.

| Memory Buffer | Prover Model | Accuracy (%) MATH | Accuracy (%) AIME |
|:---:|:---:|:---:|:---:|
| ✗ | ✗ | 93.0 | 50.0 |
| ✗ | ✓ | 97.0 | 60.0 |
| ✓ | ✗ | 93.0 | 50.0 |
| ✓ | ✓ | **97.4** | **66.7** |

*Table 3.* Accuracy (%) of DeepSeek-V3.1 with *Hermes* on HM2 with a combination of various autoformalizer sampling budget ($K_f$) and prover sampling budget ($K_p$).

| | | Prover (@$K_p$) | | | |
|:---:|:---:|:---:|:---:|:---:|:---:|
| | | @1 | @4 | @8 | @16 |
| Autoformalizer | @1 | 15.6 | 16.6 | 15.2 | 16.6 |
| (@$K_f$) | @4 | 23.7 | 30.3 | 30.8 | 31.8 |
| | @8 | 23.2 | 30.8 | 31.8 | 33.7 |

come more pronounced as reasoning complexity increases.

### 5.6. Sensitivity analysis on various values of $K_p$ and $K_f$

Table 3 illustrates the sensitivity of *Hermes* to sampling hyperparameters. We vary the autoformalizer sampling budget ($K_f$) from 1 to 8 and prover sampling budget ($K_p$) from 1 to 16, which is the number of formalization and proof verification attempts, respectively. In this experiment, we use DeepSeek-V3.1 as the reasoning model and conduct evaluations on the HARDMath2. Overall, increasing either $K_f$ or $K_p$, or both jointly, leads to improved performance. Ta-

ble 3 shows that increasing $K_f$ from 1 to 4 yields a massive 82.2% accuracy boost by mitigating formalization errors, with diminishing returns beyond that. Similarly, raising $K_p$ to 4 improves accuracy by 22.3% through better proof exploration, also saturating at higher values. We therefore select $K_f$=4 and $K_p$=4 to balance reliability with computational cost. Additional ablations are in Appendix B.

### 5.7. Translation accuracy and verification coverage

To evaluate the reliability of *Hermes*, we performed a human-led audit of the formalized reasoning steps. In Ta-

ble 4, we report two primary metrics: *CoT Coverage*, representing the proportion of formalized steps for which the Prover Module successfully identified a proof, and *Translation Accuracy*, defined as the proportion of steps where human experts verified the formalization as semantically faithful to the original statement. Adopting the annotation protocols by (Gao et al., 2025a), we manually labeled 100 steps: 50 randomly selected successful attempts (Table 5) and 50 steps where the prover failed to find a proof (Table 6). We also measured the density of tool calls to understand how *Hermes* is utilized across different difficulty levels. We found that DeepSeek-V3.2 calls the module 5.5 times on MATH and 10.4 times on AIME, respectively. Viewed alongside our 75% coverage rate, this trend suggests that for more challenging tasks, the model more explicitly relies on *Hermes* to break down the reasoning chain into a more extensive series of sub-steps. Furthermore, qualitative inspection of the Formalizer Module's output confirms that over 80% of these steps are faithfully translated into Lean.

*Table 4.* CoT coverage and translation accuracy of *Hermes* based on the DeepSeek-V3.2 outputs.

| Metric | MATH | AIME |
| --- | --- | --- |
| CoT Coverage | 87% | 76% |
| Translation Accuracy | 86% | 80% |

To further refine this analysis, we categorized the manually annotated steps by mathematical domain, as shown in Table 5. We observe that the agent's performance is strongest in equations, polynomials, and algebra. This high performance correlates directly with the density of existing declarations in Mathlib. According to the Lean Community (The Lean Community, 2026), algebra alone contains $56,000+$ declarations, whereas Geometry ($\sim 5800$), Number Theory ($\sim 6500$), and Combinatorics ($\sim 7500$) are significantly less saturated.

*Table 5.* Topic distribution of correctly formalized steps based on the DeepSeek-V3.2 outputs.

| Topic | MATH | AIME |
| --- | --- | --- |
| equations/polynomials | 21 | 16 |
| arithmetic | 12 | 6 |
| inequality | 5 | 3 |
| geometry | 1 | 6 |
| sets/sequences | 3 | 4 |
| trigonometric equations | 2 | 3 |
| functions | 3 | 2 |
| combinatorics | 0 | 9 |
| others | 3 | 1 |

Finally, to diagnose failure modes within *Hermes*, we classified a subset of steps for which the Prover Module could find

neither a valid proof nor a counter-proof (Table 6). These results underscore the agent's dependency on the current state of the formal mathematical library. However, because *Hermes* is designed with modularity in mind, its performance will improve with the ATP community as better prover and autoformalization models emerge.

*Table 6.* Topic distribution of formalized steps with no found proof based on the DeepSeek-V3.2 outputs.

| Topic | MATH | AIME |
| --- | --- | --- |
| geometry | 16 | 14 |
| trigonometry | 6 | 5 |
| functions | 7 | 3 |
| number theory | 0 | 8 |
| sets/sequences | 4 | 9 |
| equations/polynomials | 7 | 6 |
| others | 10 | 5 |

## 6. Conclusion

In this work, we introduced *Hermes*, a Lean4-powered, tool-augmented agent for advanced mathematical reasoning. The framework combines the flexibility of informal problem solving with the rigor of formal verification by validating intermediate steps that would otherwise be prone to hallucinations or logical errors. Built on state-of-the-art Lean4 theorem-proving and autoformalization models, *Hermes* incorporates a dedicated memory mechanism that preserves proof continuity across long reasoning chains and guides the reasoning LLM towards a correct solution. Experimental results show that *Hermes* achieves higher accuracy on four challenging mathematical benchmarks while consuming substantially fewer inference resources than reward-based approaches. On average, *Hermes* improves accuracy by 14% while using at least $4\times$ fewer reasoning tokens. This improvement stems from the synergistic interaction between its memory module, back-translator, and Lean4's native solvers. Overall, *Hermes* represents a step toward unifying informal and formal mathematical reasoning within a scalable, verifiable, interpretable, and modular framework.

## Acknowledgments

The work of Farzan Farnia is partially supported by a grant from the Research Grants Council of the Hong Kong Special Administrative Region, China, Project 14210725, and is partially supported by CUHK Direct Research Grants with CUHK Project No. 4055164. The work is also supported by a grant under 1+1+1 CUHK-CUHK(SZ)-GDSTC Joint Collaboration Fund. The authors also thank the anonymous reviewers for their helpful feedback and constructive suggestions.

## Impact Statement

This paper introduces a hybrid agent designed to bridge the gap between informal and formal mathematics. The research relies exclusively on publicly available mathematical data and does not involve human subjects or sensitive personal information; as such, it poses no privacy or fairness concerns. By integrating Lean into informal mathematics via tool-based agent, this work aims to enhance automated theorem proving. We confirm that this research adheres strictly to the ICML Code of Ethics and presents no foreseeable harmful applications.

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

## A. Prompt and instruction templates

```
# Hermes Tool Instruction
Formally validates a **single** reasoning step using a formal Lean4 verifier. Invoke this
    function when facing a step that potentially involves a hallucination. Make sure to
    verify EVERY critical mathematical proof step.

Args:
    proof_step (str): A proof step that includes both the goal to be proven and the
    relevant context (e.g., variables, assumptions, and previously proven statements).
    Always explicitly specify the relevant context, such as domains, data types, and any
    other necessary details. Make sure to state the proof step in English.

Returns:
    str: A status string indicating the verification result:
        - **CORRECT**: Step verified by the theorem prover.
        - **INCORRECT**: Step rejected by the theorem prover (e.g., the prover proved a
    contradiction or the opposite statement).
        - **VERIFICATION FAILURE**: Step could not be verified by Lean 4 (e.g., the prover
    was unable to prove the statement or find contradictory arguments)

Notes:
    - Treat **CORRECT** steps as reliable within the given formalization.
    - Treat **INCORRECT** steps as requiring revision; a suggested correction is returned
    with this label.
    - Use **VERIFICATION FAILURE** to indicate inconclusive or ill-formed steps that Lean 4
    was unable to prove or disprove.
```

```
# Answer verification prompt
Given a question, an answer to that question, and the ground truth for that question's
    answer, you need to check if the given answer matches the ground truth.
* If the answer is complete and correct, simply reply with True.
* If the given answer does not match the ground truth or is incomplete, reply with False.
* Do **NOT** respond with any other characters.

Question:
<question>

Answer:
<answer>

Ground Truth:
<ground_truth>

Does the answer match the ground truth? (True or False):
```

```
# NL->FL translation prompt for Goedel-Formalizer-V2-8B
Please autoformalize the following natural language problem statement in Lean 4.
Use the following theorem name: test
The natural language statement is:
<question>
Think before you provide the lean statement.
```

```
# Theorem proving prompt for Goedel-Prover-V2-8B
Complete the following Lean 4 code:

```lean4
<header>

<body>```

Before producing the Lean 4 code to formally prove the given theorem, provide a detailed
    proof plan outlining the main proof steps and strategies.
The plan should highlight key ideas, intermediate lemmas, and proof structures that will
    guide the construction of the final formal proof.
```

```
# NL->FL translation prompt for Kimina-Autoformalizer-7B
Please autoformalize the following problem in Lean 4 with a header. Use the following
    theorem names: my_favorite_theorem.

<question>
```

```
# Theorem proving prompt for Kimina-Prover-RL-1.7B
Think about and solve the following problem step by step in Lean 4.
# Formal statement:
```lean4
<header>

<body>
```
```

The prompt instructions and templates are shown above. For the *Hermes* prompt, we follow the LangChain community guidelines. The tool is named `verify_one_mathematical_step` to encourage the model to verify only a single step rather than an entire proof. The tool description is flexible and can be modified; in particular, adjusting the description can change how frequently the tool is invoked. The answer-verification prompt is adapted from (Liu et al., 2025). For the autoformalizer and prover prompts, we use the officially released versions provided by the original developers.

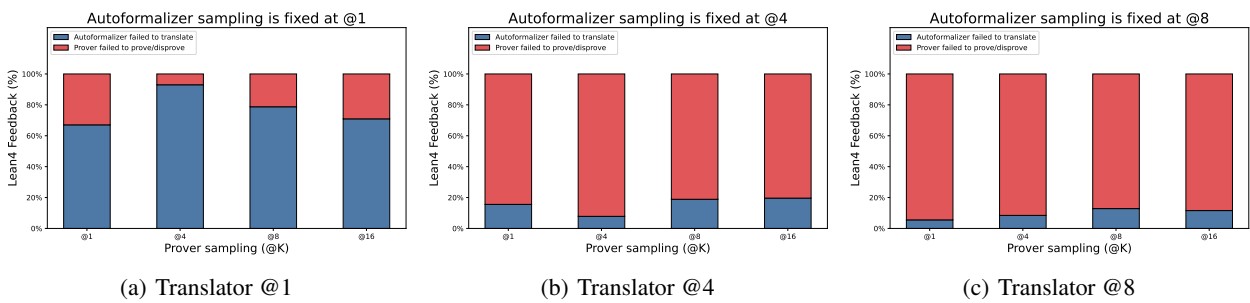

*Figure 6.* Distribution of *Hermes* failures with a fixed translation budget ($K_f$).

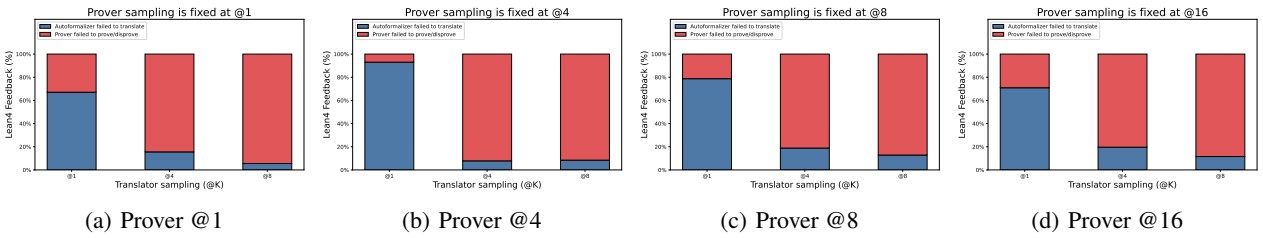

*Figure 7.* Distribution of *Hermes* failures with a fixed prover budget ($K_p$).

## B. Additional ablation on different combinations of autoformalizers and provers

Following the ablation studies presented in Table 3 of the main text, we provide additional insights into the selection of $K_f$ and $K_p$. Figures 6 and 7 illustrate the proportion of translator and prover errors, respectively, each with either the translation or prover budget held fixed. We observe that at @1, translation failures are overwhelmingly frequent, exceeding 60%. As the translation budget $K_f$ increases, such failures become rare, contributing to 10% or less of all Lean verification errors. Conversely, when $K_p = 1$, the prover becomes highly unreliable, with unresolved proof steps accounting for up to 95% of all verification failures. Increasing $K_p$ to 16 yields diminishing returns, as theorem proving is inherently more complex than autoformalization, requiring a substantially larger budget for high-quality proof generation. However, given that *Hermes* was evaluated under moderate computational constraints, our focus remained on efficient inference and agent interaction. Our results align with prior work, where whole-proof generation models typically employ very large sampling budgets, ranging from 1,024 to 25,600 samples. We hypothesize that in large-scale settings, increasing $K_p$ beyond 16 may further reduce prover failure rates.

**[MATH500] Precalculus - 768**

Find all solutions to $\sin\left(\tan^{-1}(x) + \cot^{-1}\left(\frac{1}{x}\right)\right) = \frac{1}{3}$. Enter all the solutions, separated by commas.

---

...
#### Case 1: \(x > 0\)
If \(x > 0\), then \(y = \frac{1}{x} > 0\). Therefore, the identity holds:
\[ \cot^{-1}\left( \frac{1}{x} \right) = \tan^{-1}(x) \]

...

Thus, both solutions are valid.

#### Case 2: \(x < 0\)
If \(x < 0\), then \(y = \frac{1}{x} < 0\). The identity \(\cot^{-1}(y) = \tan^{-1}\left( \frac{1}{y} \right)\) doesn't hold directly.

I recall that for \(y < 0\), the following identity is true:
\[ \cot^{-1}(y) = \pi + \tan^{-1}\left( \frac{1}{y} \right) \]

But, I'm not sure if this is accurate. Let me verify.

Looking at the range of \(\cot^{-1}(y)\):
- The range of \(\cot^{-1}(y)\) is \((0, \pi)\).
- For \(y > 0\), \(\cot^{-1}(y) = \tan^{-1}(1/y)\).
- For \(y < 0\), \(\cot^{-1}(y) = \pi + \tan^{-1}(1/y)\).

But, \(\tan^{-1}(1/y)\) for \(y < 0\) will give a value between \(-\pi/2\) and \(0\). Adding \(\pi\) to it gives a value between \(\pi/2\) and \(\pi\), which is within the range of \(\cot^{-1}(y)\).

Thus, for \(x < 0\):
\[ \cot^{-1}\left( \frac{1}{x} \right) = \pi + \tan^{-1}(x) \]

Now, the expression inside the sine becomes:
\[ \tan^{-1}(x) + \cot^{-1}\left( \frac{1}{x} \right) = \tan^{-1}(x) + \pi + \tan^{-1}(x) = \pi + 2\tan^{-1}(x) \]

Thus, the equation is:
\[ \sin(\pi + 2\tan^{-1}(x)) = \frac{1}{3} \]

I know that \(\sin(\pi + \theta) = -\sin(\theta)\). Therefore:
\[ \sin(\pi + 2\tan^{-1}(x)) = -\sin(2\tan^{-1}(x)) = \frac{1}{3} \]

But from the previous case, \(\sin(2\tan^{-1}(x)) = \frac{2x}{1 + x^2}\). Therefore:
\[ -\frac{2x}{1 + x^2} = \frac{1}{3} \]

Simplify:
\[ \frac{-2x}{1 + x^2} = \frac{1}{3} \]
\[ -6x = 1 + x^2 \]
\[ x^2 + 6x + 1 = 0 \]

The solutions are:
\[ x = \frac{-6 \pm \sqrt{36 - 4}}{2} = \frac{-6 \pm \sqrt{32}}{2} = \frac{-6 \pm 4\sqrt{2}}{2} = -3 \pm 2\sqrt{2} \]

Now, since \(x < 0\) in this case, we need to check both solutions:
1. \(x = -3 + 2\sqrt{2} \approx -3 + 2.828 = -0.172 < 0\)
2. \(x = -3 - 2\sqrt{2} \approx -3 - 2.828 = -5.828 < 0\)

Both solutions are negative, so both are valid.

...
### Summary of Solutions
From both cases, we have four potential solutions:
1. \(x = 3 + 2\sqrt{2}\)
2. \(x = 3 - 2\sqrt{2}\)
3. \(x = -3 + 2\sqrt{2}\)
4. \(x = -3 - 2\sqrt{2}\)

All of them satisfy the original equation, as verified.

### Final Answer
The solutions are \(\boxed{3 + 2\sqrt{2}}\), \(\boxed{3 - 2\sqrt{2}}\), \(\boxed{-3 + 2\sqrt{2}}\), and \(\boxed{-3 - 2\sqrt{2}}\).

---

...
First, I notice that $\tan^{-1}(x)$ and $\cot^{-1}\left(\frac{1}{x}\right)$ are related. Let me verify if there's a relationship between these two inverse trigonometric functions.

*Lean agent verified proof steps:*
  1. *For $x > 0$, $\theta = tan^{-1}(x)$ implies $tan(\theta) = x$, so $cot(\theta) = 1/x$, which means $\theta = cot^{-1}(1/x)$. Therefore, $tan^{-1}(x) = cot^{-1}(1/x)$ for $x > 0$.*
  2. *For $x = 0$, $tan^{-1}(0) = 0$ and $cot^{-1}(1/0)$ is undefined since division by zero is not allowed. Therefore, $x = 0$ is not in the domain.*

Let me continue the analysis:

From the verification, I can see that for x ≠ 0, tan⁻¹(x) = cot⁻¹(1/x). This means the expression simplifies to:

sin(2tan⁻¹(x)) = 1/3

Let me continue the mathematical reasoning:

Let θ = tan⁻¹(x), so x = tan(θ). Then we have:
sin(2θ) = 1/3

Using the identity sin(2θ) = 2tan(θ)/(1 + tan²(θ)) = 2x/(1 + x²)

So the equation becomes:
2x/(1 + x²) = 1/3

Solving this:
6x = 1 + x²
x² - 6x + 1 = 0

Using the quadratic formula:
x = [6 ± √(36 - 4)]/2 = [6 ± √32]/2 = [6 ± 4√2]/2 = 3 ± 2√2

Now I need to check if both solutions are valid. Let me verify if they satisfy the original equation:

*Lean verified proof steps:*
  1. *For $x = 3 + 2\sqrt{2}$, $tan^{-1}(x)$ is defined since $x > 0$, and $cot^{-1}(1/x)$ is also defined since $1/x > 0$. The sum $tan^{-1}(x) + cot^{-1}(1/x) = 2tan^{-1}(x)$ is valid, and $sin(2tan^{-1}(x)) = 2x/(1+x²) = 1/3$, so this is a valid solution.*
  2. *For $x = 3 - 2\sqrt{2}$, since $3 - 2\sqrt{2} \approx 3 - 2.828 = 0.172 > 0$, $tan^{-1}(x)$ is defined and $cot^{-1}(1/x)$ is also defined. The sum $tan^{-1}(x) + cot^{-1}(1/x) = 2tan^{-1}(x)$ is valid, and $sin(2tan^{-1}(x)) = 2x/(1+x²) = 1/3$, so this is also a valid solution.*

Therefore, both solutions are valid.

**Final Answer:** 3 + 2√2, 3 - 2√2

---

*Figure 8.* Illustration of *Hermes*'s effect on chain-of-thought generation: the model attempts to solve the precalculus-768 problem from the MATH500 dataset, and *Hermes* detects and corrects a hallucinated reasoning step.

## C. Example of a *Hermes* corrected problem

Figure 8 shows a case where *Hermes* helps the LLM avoid pursuing an incorrect reasoning path, in this example, attempting to prove that $\tan^{-1}(x) = \cot^{-1}(1/x)$ for $x < 0$. The Lean-validated steps guide the model away from invalid proof directions. We observe that LLMs often generate answers with unwarranted confidence and without verifying intermediate steps. Our agent mitigates this behavior by enforcing step-by-step validation and stricter reasoning discipline.

## D. Additional formalization examples across MATH and AIME datasets

Figures 9–11 and 12-14 show qualitative examples of formalized reasoning steps from the MATH500 and AIME datasets, respectively. Each example pairs an informal natural-language step with its corresponding formal statement in Lean, illustrating the kinds of arithmetic, combinatorial, and number-theoretic reasoning steps covered by our benchmark.

**Informal Step:** Given that: $-f(f(1)) = 4. - f(1) = 2$ when $f(x) = 2^x$. Prove that: $f(f(f(1))) = f(4) = 2^4 = 16$.

```
/-Formal Statement-/
theorem test (f : ℕ → ℕ) (h1 : f (f 1) = 4) (h2 : ∀ x, f x = 2^x) : f (f (f 1)) = 16
    ∧ f 4 = 16 := by
  sorry
```

*Figure 9.* Example #1 of a MATH500 formalized step

**Informal Step:** Given 5 shirts, 6 pairs of pants, and 8 hats, the total number of outfits (one of each) is $5 \times 6 \times 8 = 240$.

```
/-Formal Statement-/
theorem test : let shirts := Finset.range 5 let pants := Finset.range 6 let hats :=
    Finset.range 8 (shirts × pants × hats).card = 240 := by
  sorry
```

*Figure 10.* Example #2 of a MATH500 formalized step

**Informal Step:** Given that $24^{(24^24)}$ is even, and the pattern shows that $24^n \equiv 6 \mod 10$ when $n$ is even, therefore $24^{(24^{(24^{24})})} \equiv 6 \mod 10$.

```
/-Formal Statement-/
theorem test : (Even (24^(24^24))) → (∀ n : ℕ, Even n → 24^n ≡ 6 [MOD 10]) →
    (24^(24^(24^24)) ≡ 6 [MOD 10]) := by
  sorry
```

*Figure 11.* Example #3 of a MATH500 formalized step

**Informal Step:** Given a positive integer $N = 3^4 \times 5^2$, the set A of positive integer divisors of $N$ consists of all numbers of the form $3^a \times 5^b$ where $0 \leq a \leq 4$ and $0 \leq b \leq 2$. The number of such divisors is $(4+1) \times (2+1) = 15$.

```
/-Formal Statement-/
theorem test : let N := 3^4 * 5^2 let A := {d : ℕ | ∃ a b, a \leq 4 ∧ b \leq 2 ∧ d =
    3^a * 5^b} (∀ x, x ∈ Nat.divisors N ↔ x ∈ A) ∧ ((Nat.divisors N).card = (4 + 1) *
    (2 + 1)) ∧ ((4 + 1) * (2 + 1) = 15) := by
  sorry
```

*Figure 12.* Example #1 of an AIME formalized step

**Informal Step:** Given that: If $d$ divides $3(d + 1)(d^2 - 4d + 13)$ and $d$ is coprime to $d + 1$, then $d$ divides $3(d^2 - 4d + 13)$. Prove that: The positive integers $n$ such that $n + 2$ divides $3(n + 3)(n^2 + 9)$ are $n = 1, 11$, and 37. Their sum is $1 + 11 + 37 = 49$.

```
/-Formal Statement-/
theorem test : (∀ d : ℤ, d | 3 * (d + 1) * (d^2 - 4*d + 13) → Int.gcd d (d + 1) = 1 →
    d | 3 * (d^2 - 4*d + 13)) → {n : ℕ | n > 0 ∧ (n + 2 : ℤ) | 3 * (n + 3) * (n^2 +
    9)} = {1, 11, 37} ∧ 1 + 11 + 37 = 49 := by
  sorry
```

*Figure 13.* Example #2 of an AIME formalized step

**Informal Step:** Prove that: Given that the total number of ways is $9! \times 56 \times (3!)^6$, we compute the prime factorization: $9! = 2^7 \times 3^4 \times 5 \times 7$; $56 = 2^3 \times 7$; $(3!)^6 = 2^6 \times 3^6$. Multiplying yields $2^{(7+3+6)} \times 3^{(4+6)} \times 5^1 \times 7^{(1+1)} = 2^16 \times 3^10 \times 5^1 \times 7^2$.

```
/-Formal Statement-/
theorem test : 9! = 2^7 * 3^4 * 5 * 7 ∧ 56 = 2^3 * 7 ∧ (3!)^6 = 2^6 * 3^6 ∧ 9! * 56 *
    (3!)^6 = 2^16 * 3^10 * 5 * 7^2 := by
  sorry
```

*Figure 14.* Example #3 of an AIME formalized step

*Table 7.* Mean accuracy (%) $\pm$ standard deviation of various models under four inference strategies: zero-shot CoT (@1), majority vote (@5), reward-model selection (Best-of-5), and *Hermes* (@1). Results are averaged over five independent runs on two benchmarks: MATH500 and AIME'25

| | Qwen3-8B | | OpenAI o3-mini | | DeepSeek-V3.1 | | DeepSeek-V3.2 | |
|---|---|---|---|---|---|---|---|---|
| | MATH500 | AIME'25 | MATH500 | AIME'25 | MATH500 | AIME'25 | MATH500 | AIME'25 |
| CoT@1 | $85.7 \pm 0.7$ | $18.7 \pm 2.7$ | $96.2 \pm 0.3$ | $71.3 \pm 5.0$ | $95.6 \pm 0.7$ | $48.0 \pm 3.4$ | $96.6 \pm 0.6$ | $49.3 \pm 3.9$ |
| CoT@5+Majority | $86.8 \pm 0.3$ | $20.0 \pm 2.1$ | $96.5 \pm 0.3$ | $64.7 \pm 3.4$ | $96.9 \pm 0.2$ | $48.0 \pm 3.4$ | $97.2 \pm 0.5$ | $58.7 \pm 5.8$ |
| CoT@5+Skywork | $90.9 \pm 0.2$ | $30.7 \pm 2.5$ | $96.4 \pm 0.4$ | $72.0 \pm 5.8$ | $96.7 \pm 0.1$ | $52.0 \pm 2.7$ | $97.3 \pm 0.1$ | $60.0 \pm 4.7$ |
| CoT@5+ArmoRM | $89.2 \pm 0.5$ | $30.7 \pm 2.5$ | $96.1 \pm 0.5$ | $72.0 \pm 5.8$ | $96.4 \pm 0.5$ | $55.3 \pm 1.6$ | $97.0 \pm 0.2$ | $55.3 \pm 1.6$ |
| CoT@5+Shepherd | $86.9 \pm 0.7$ | $25.3 \pm 3.4$ | $96.1 \pm 0.2$ | $70.0 \pm 7.0$ | $96.4 \pm 0.4$ | $57.3 \pm 5.7$ | $96.7 \pm 0.3$ | $57.3 \pm 5.7$ |
| CoT@5+RLHFlow | $83.6 \pm 0.3$ | $24.0 \pm 3.9$ | $95.8 \pm 0.4$ | $65.3 \pm 4.0$ | $96.0 \pm 0.7$ | $50.0 \pm 2.1$ | $95.5 \pm 0.2$ | $53.3 \pm 3.7$ |
| *Lean-based methods* | | | | | | | | |
| CoT@5+Safe | $89.1 \pm 0.3$ | $22.0 \pm 1.6$ | $95.5 \pm 0.3$ | $80.0 \pm 3.0$ | $96.0 \pm 0.3$ | $45.3 \pm 2.7$ | $97.1 \pm 0.2$ | $46.7 \pm 2.1$ |
| CoT@5+Safe* | $89.2 \pm 0.4$ | $21.3 \pm 3.4$ | $96.4 \pm 0.4$ | $80.7 \pm 2.5$ | $96.4 \pm 0.5$ | $50.0 \pm 3.7$ | $97.3 \pm 0.2$ | $52.7 \pm 2.5$ |
| ***Hermes*@1** | $91.3 \pm 1.1$ | $28.0 \pm 1.6$ | $97.0 \pm 0.7$ | $84.0 \pm 2.5$ | $97.2 \pm 0.5$ | $66.7 \pm 3.0$ | $98.2 \pm 0.4$ | $74.7 \pm 5.4$ |

# E. Additional results on benchmark results

Complementing Table 1, we report the mean and standard deviation for all inference methods on the MATH500 and AIME'25 datasets, based on five independent runs. We observe that the performance of *Hermes* is stable, with variance comparable to that of CoT and RM-augmented baselines.

# F. *Hermes* Algorithm

---

**Algorithm 1** Hermes Pseudocode

---

**Input:** Step $s_t$, Memory Database $\mathcal{D}$, Hyperparameters $K_f, K_p, k$
**Output:** Tuple $(s_t, \text{status}, \mathcal{D})$

$C \leftarrow \text{RETRIEVETOPK}(\mathcal{D}, s_t, k)$      ▷ *Retrieve top-k relevant correct steps from database*
$S \leftarrow s_t \cup C$      ▷ *Augment context with retrieval*
$\text{status} \leftarrow \text{VERIFICATION FAILURE}$

**for** $i = 1$ **to** $K_f$ **do**
    $f_t \leftarrow \pi_f(S)$      ▷ *Translate to Lean*
    **if** $\text{REPL}(f_t)$ is True **then**
        $\tilde{s}_t \leftarrow \text{BackTranslation}(f_t)$
        **if** $s_t \approx \tilde{s}_t$ **then**
            **break**      ▷ *Exit Formalization Module only when both REPL and Backtranslation are passed*
        **end if**
    **end if**
**end for**
**if** $f_t = \emptyset$ **then**
    **Return** $(s_t, \text{status}, \mathcal{D})$      ▷ *Verification failure is returned if no translation passed REPL and backtranslation*
**end if**

**for** $j = 1$ **to** $K_p$ **do**
    $p_t^+ \leftarrow \pi_p(\cdot | f_t)$      ▷ *Proof for goal*
    $p_t^- \leftarrow \pi_p(\cdot | \neg f_t)$      ▷ *Proof for counter-goal*
    **if** $\text{REPL}(p_t^+)$ is True **then**
        $\text{status} \leftarrow \text{CORRECTSTEP}$
        $\mathcal{D} \leftarrow \mathcal{D} \cup \{s_t\}$      ▷ *Update database*
        **break**
    **else if** $\text{REPL}(p_t^-)$ is True **then**
        $\text{status} \leftarrow \text{REVISESTEP}$
        **break**
    **end if**
**end for**
**Return** $(s_t, \text{status}, \mathcal{D})$

---

The *Hermes* algorithm (Algorithm 1) takes a natural language reasoning step $s_t$, a memory database $\mathcal{D}$, hyperparameters $K_f, K_p$ for sampling formalizations and proofs, and retrieval parameter $k$. It begins by retrieving the top-$k$ relevant, verified steps from $\mathcal{D}$ using cosine similarity of vectors that represent embedded proof steps to augment the context $S$. The algorithm then attempts to formalize the step into Lean code ($f_t$) using a formalizer policy $\pi_f$, repeating up to $K_f$ times until a candidate passes both REPL verification and semantic consistency checks via back-translation. If formalization fails, the step is marked as a verification failure. Otherwise, the algorithm proceeds to the prover module, which attempts to generate a valid Lean4 proof for either the initial goal ($p_t^+$) or its negation ($p_t^-$) up to $K_p$ times. If a proof for the statement is found, the step is marked as correct and added to the database $\mathcal{D}$; if a counter-example/proof is found, the step is flagged and LLM is instructed to explore other proof steps; if neither goal was proved, the agent returns verification failure signal back to the reasoning LLM.

## G. Distribution of successfully proved problems by math topic under different inference strategies

To better understand the impact of our agent on the underlying reasoning models, we analyze the distribution of solved problems across different mathematical topics from MATH500 (Tables 8, 9, 10, 11) and HardMath2 (Tables 12, 13, 14, 15). Our results indicate that most mathematical domains benefit from the inclusion of Lean-based verification, although a few topics experience slight performance drops. Geometry, for instance, shows lower gains, which we attribute to the limited availability of geometry-related theorems in Mathlib, the primary mathematical library for Lean. Furthermore, we observe that certain reward models exhibit domain-specific strengths: for example, Skywork consistently performs better in geometry-related tasks. Overall, we expect that as Lean and Mathlib continue to evolve, these performance gaps will diminish, and the verification capabilities of our agent will further improve with future updates to theorem provers and supporting libraries.

*Table 8.* Distribution of correct answers by topic on MATH500 for the DeepSeek-V3.1 base reasoning model.

| | Algebra | Counting & Probability | Geometry | Intermediate Algebra | Number Theory | Pre-algebra | Pre-calculus |
|---|---|---|---|---|---|---|---|
| ZS-CoT | 123 | 33 | 35 | 91 | 62 | 78 | 52 |
| *Hermes* | **124** | **36** | 36 | **93** | 62 | **80** | **56** |
| Skywork | 123 | 35 | **39** | 92 | 62 | 77 | 55 |
| ArmoRM | 123 | 35 | **39** | 89 | 62 | 76 | 54 |
| Shepherd | 123 | **36** | 38 | 92 | 62 | 77 | 53 |
| RLHFlow | 123 | 34 | 35 | 91 | 62 | 79 | 53 |

*Table 9.* Distribution of correct answers by topic on MATH500 for the DeepSeek-V3.2 base reasoning model.

| | Algebra | Counting & Probability | Geometry | Intermediate Algebra | Number Theory | Pre-algebra | Pre-calculus |
|---|---|---|---|---|---|---|---|
| ZS-CoT | 124 | 36 | 34 | 93 | 61 | 81 | 53 |
| Hermes | 123 | 36 | 39 | 96 | 61 | 81 | 55 |
| Skywork | 124 | 36 | 35 | 96 | 61 | 80 | 55 |
| ArmoRM | 124 | 36 | 36 | 94 | 61 | 80 | 55 |
| Shepherd | 124 | 36 | 35 | 95 | 62 | 78 | 55 |
| RLHFlow | 124 | 36 | 36 | 95 | 62 | 80 | 53 |

*Table 10.* Distribution of correct answers by topic on MATH500 for the o3-mini base reasoning model.

| | Algebra | Counting & Probability | Geometry | Intermediate Algebra | Number Theory | Pre-algebra | Pre-calculus |
|---|---|---|---|---|---|---|---|
| ZS-CoT | 124 | **38** | 35 | 89 | 62 | 77 | 54 |
| *Hermes* | 124 | 37 | **37** | 93 | 61 | **79** | 55 |
| Skywork | 123 | 37 | 36 | **94** | 62 | 77 | 55 |
| ArmoRM | 124 | 37 | 35 | 93 | 61 | 76 | 55 |
| Shepherd | 124 | **38** | 35 | 91 | 62 | 78 | 54 |
| RLHFlow | 123 | 37 | 36 | 91 | 61 | 78 | 53 |

*Table 11.* Distribution of correct answers by topic on MATH500 for the Qwen3-8B base reasoning model.

|          | Algebra | Counting & Probability | Geometry | Intermediate Algebra | Number Theory | Pre-algebra | Pre-calculus |
|----------|---------|------------------------|----------|----------------------|---------------|-------------|--------------|
| ZS-CoT   | 120     | 28                     | 27       | 70                   | 60            | 75          | 44           |
| *Hermes* | **123** | **33**                 | 31       | **84**               | 61            | **77**      | 47           |
| Skywork  | 120     | 32                     | **32**   | 82                   | **62**        | **77**      | **50**       |
| ArmoRM   | 118     | 31                     | 31       | 80                   | **62**        | 75          | 46           |
| Shepherd | 121     | 31                     | 29       | 72                   | 61            | **77**      | 48           |
| RLHFlow  | 118     | 28                     | 23       | 72                   | 58            | 73          | 48           |

*Table 12.* Distribution of correct answers by topic on HARDMath2 for the DeepSeek-V3.1 base reasoning model.

|          | asympytotic series | boundary layers | integrals | nonlinear ode | nonlinear pde | other | wkb |
|----------|--------------------|-----------------|-----------|---------------|---------------|-------|-----|
| ZS-CoT   | 0                  | 22              | 9         | 2             | 8             | 3     | 4   |
| *Hermes* | 0                  | **28**          | **10**    | 3             | 14            | 3     | 6   |
| Skywork  | 0                  | 23              | 8         | 3             | 16            | 3     | 7   |
| ArmoRM   | 0                  | 21              | **10**    | 3             | 12            | 3     | 7   |
| Shepherd | 0                  | 21              | 9         | 2             | **17**        | 2     | 8   |
| RLHFlow  | 0                  | 20              | 8         | 1             | **17**        | 1     | 7   |

*Table 13.* Distribution of correct answers by topic on HARDMath2 for the DeepSeek-V3.2 base reasoning model.

|          | asympytotic series | boundary layers | integrals | nonlinear ode | nonlinear pde | other | wkb |
|----------|--------------------|-----------------|-----------|---------------|---------------|-------|-----|
| ZS-CoT   | 0                  | 23              | 8         | 2             | 13            | 3     | 9   |
| Hermes   | 0                  | 28              | 10        | 3             | 14            | 3     | 6   |
| Skywork  | 0                  | 25              | 10        | 1             | 17            | 3     | 9   |
| ArmoRM   | 0                  | 23              | 8         | 2             | 15            | 3     | 9   |
| Shepherd | 0                  | 23              | 7         | 2             | 15            | 3     | 9   |
| RLHFlow  | 0                  | 21              | 6         | 3             | 14            | 2     | 6   |

*Table 14.* Distribution of correct answers by topic on HARDMath2 for the o3-mini base reasoning model.

|  | asympytotic series | boundary layers | integrals | nonlinear ode | nonlinear pde | other | wkb |
|---|---|---|---|---|---|---|---|
| ZS-CoT | 0 | 14 | 10 | 3 | 13 | 1 | 8 |
| *Hermes* | 1 | **25** | 10 | 3 | 16 | 1 | **10** |
| Skywork | 1 | 22 | 10 | 1 | **19** | 1 | 8 |
| ArmoRM | 1 | 21 | 11 | 3 | 16 | 1 | 7 |
| Shepherd | 0 | 12 | **12** | 3 | 17 | 1 | 9 |
| RLHFlow | **2** | 21 | 10 | 2 | 16 | 1 | 8 |

*Table 15.* Distribution of correct answers by topic on HARDMath2 for the Qwen3-8B base reasoning model.

|  | asympytotic series | boundary layers | integrals | nonlinear ode | nonlinear pde | other | wkb |
|---|---|---|---|---|---|---|---|
| ZS-CoT | 0 | 0 | 6 | 3 | 0 | 0 | 0 |
| *Hermes* | 0 | **2** | **8** | 2 | 0 | 0 | 2 |
| Skywork | 0 | 0 | 7 | 3 | 0 | 0 | 2 |
| ArmoRM | 0 | 0 | 6 | 3 | 0 | 0 | 2 |
| Shepherd | 0 | 0 | 7 | 3 | 0 | 0 | 2 |
| RLHFlow | 0 | 0 | 7 | 3 | 0 | 0 | 2 |

