# OpenReview forum: "HERMES: Towards Efficient and Verifiable Mathematical Reasoning in LLMs"
_ICML.cc/2026/Conference — ICML 2026 regular_

### Official Review · Reviewer_1NYm · 2026-03-06

**Soundness:** 3
**Presentation:** 3
**Significance:** 3
**Originality:** 3
**Overall Recommendation:** 5
**Confidence:** 4

**Summary:**

This paper proposes Hermes, a tool-augmented agent that interleaves informal LLM reasoning with formal Lean4 verification during mathematical problem solving. The system comprises four modules: a reasoning LLM that generates natural-language proof steps, a formalization module that translates selected steps into Lean4 statements (with back-translation for semantic consistency checking), a prover module that attempts to prove or disprove the formalized goal, and a feedback module that routes the verification signal back to the LLM. Additionally, a memory block stores previously verified steps with vector embeddings for top-k retrieval, injecting them as hypotheses into subsequent formalizations to maintain cross-step consistency. The LLM autonomously decides which "critical" steps to verify via tool-calling. Experiments on MATH-500, AIME'25, CollegeMath, and HARDMath2 across Qwen3-8B, o3-mini, and DeepSeek-V3.2 show that Hermes@1 generally outperforms CoT and Best-of-5 reward-model baselines while consuming fewer reasoning tokens. Ablations confirm the prover module contributes more than the memory block, and sensitivity analysis on sampling budgets Kf and Kp reveals diminishing returns beyond moderate values.

**Compliance With Llm Reviewing Policy:**

Affirmed.

**Final Justification:**

I have no further concerns.

**Key Questions For Authors:**

See weaknesses.

**Limitations:**

yes

**Strengths And Weaknesses:**

### Strengths

- The idea of interleaving step-level formal verification during informal reasoning generation, rather than applying verification post-hoc, is a well-motivated and natural architectural choice that enables on-the-fly error correction.
- The dual proof strategy of attempting both the goal and its negation in parallel is a clever design that provides stronger feedback than one-directional verification alone.
- The modular architecture cleanly separates concerns (formalization, proving, memory, feedback), allowing each component to be independently upgraded as better autoformalizers or provers become available.
- The experimental coverage across four models of different scales and providers, four benchmarks of varying difficulty, and multiple baseline families (ORM, PRM, Lean-based) is thorough and provides a credible picture of generalization.
- The example in Appendix C concretely demonstrates how Hermes catches a subtle mathematical error (the identity for cot⁻¹(1/x) when x < 0) that the LLM would otherwise confidently propagate, making the qualitative benefit tangible.

### Weaknesses

- The decision of which steps to verify is delegated entirely to the LLM's tool-calling behavior, yet there is no analysis of tool-call frequency, precision of step selection, or how sensitive the results are to the tool description prompt that controls invocation rates.
- The back-translation semantic equivalence check relies on an LLM judgment (st ≈ s̃t), which is itself fallible, but the paper neither evaluates the accuracy of this check nor discusses what happens when it incorrectly accepts or rejects a formalization.
- The token and FLOP comparisons between Hermes@1 and CoT@5+BoN baselines are inherently asymmetric: Hermes generates one reasoning trace plus multiple autoformalizer and prover calls, while BoN generates five full reasoning traces, so the token savings are largely a trivial consequence of single-trace versus multi-trace generation rather than a property of the verification mechanism itself.
- The paper reports geometry-related topics receive lower gains due to limited Mathlib coverage, but does not quantify the per-topic failure rate of formalization or proving, making it hard to assess the practical scope of the framework beyond pure algebra and number theory.
- The paper uses DeepSeek-V3.1 in some experiments (FLOP analysis, ablations, scaling plots) and DeepSeek-V3.2 in Table 1, but does not clearly justify or discuss the inconsistency between these two model versions across sections. Moreover, there seems to be a mismatch in Figure 5 (b).

---

> ### Author Rebuttal · Authors · 2026-03-31
>
> We thank the reviewer for the constructive feedback. Please find our responses to the comments under [W]eaknesses and [Q]uestions below. All new results and clarifications will be added in the revision.
>
>
> ## W1
> To address the reviewer’s comment, we conducted an empirical analysis of the frequency of tool calls. **Hermes, on average, invokes the verifier 5.5 and 10.4 times on MATH and AIME**, respectively. It suggests that on more challenging benchmarks, the model relies more on Hermes and divides the problem into smaller sub-steps.
>
> Moreover, since Hermes is a communication between Lean and the reasoning LLM, each CoT chunk has a corresponding proof attempt in Lean. We recorded each such interaction, and if a proof (or counter-proof) was found, we marked such chunks as “verified”. According to our evaluation, the **CoT coverage of AIME sits around 76%, while on the relatively easier MATH, it reaches 87%**. The informal CoT coverage is consistently high for the two datasets we experimented on.
>
> We further discuss the distribution of Lean-attempted steps in _[W4]_.
>
> ## W2
> To address the reviewer’s concern regarding the autoformalization accuracy, we conducted a manual human review of autoformalization consistency. The implemented Formalization Module achieves a **high translation accuracy of 86% on MATH and 80% on AIME**. Although we cannot guarantee the full correctness of any LLM translation system, our tests show that the current accuracy is on par with established works on autoformalization like HERALD. We distinguish failure cases as follows:
>
> - False Positives. If an incorrect step passes the equivalence check, the Lean prover must still find a formal proof. Since wrong steps are unlikely to yield valid proofs, the prover acts as a final safeguard.
>
> - False Negatives. If a valid step is rejected, the system returns a verification failure and triggers LLM self-verification. This conservative approach ensures we only pass translations with a high validity likelihood.
>
> ## W3
> We agree with the reviewer that the comparison between the Hermes@1 and CoT@5 baselines is structurally asymmetric. However, this asymmetry highlights the advantage of our method. We want to show that, instead of generating and resampling multiple full CoTs, calling an external verifier mid-generation is a more efficient approach. Therefore, while the token savings come from single-trace versus multi-trace generation, this single-trace capability is a direct consequence of calling a Lean-based agent to verify the reasoning on the fly.
>
> ## W4
> To address the reviewer’s request for topic-specific analysis, we provide a detailed report on what the model decides to formalize (Table A) and where verification most often fails (Table B).
>
> Table A shows that the model primarily invokes Lean to verify symbolically dense operations like equations and arithmetic. However, Table B reveals that fields such as geometry and combinatorics are the primary culprits behind "verification failure".
>
> Our findings also correlate with the current state of Mathlib4, where the distribution of declarations is heavily skewed. As of recent community documentation [1], Algebra contains over 56000 declarations, whereas Geometry (\~5800), Number Theory (\~6500), and Combinatorics (\~7500) are significantly less saturated. This imbalance explains the lack of performance in geometry-heavy problems; the agent attempts to formalize the reasoning, but the library may lack the specific lemmas required to close the proof.
>
> [1] https://leanprover-community.github.io//mathlib4_docs/mathlib.html
>
> We emphasize that Hermes is built to be modular and version-agnostic. It does not rely on hard-coded tactics or specific library versions. As Mathlib4 continues its rapid expansion, particularly in undersaturated fields, Hermes can be updated to utilize new declarations without architectural changes. This ensures the practical scope of the framework will expand naturally alongside the formal mathematics library saturation. We will include these tables and the associated failure analysis in the revised manuscript to clarify the current limitations and future scalability of the system.
>
> Table A: Topic Distribution of Formalized Steps
> |Topic|MATH|AIME|
> |---|---|---|
> |equations/polynomials|21|16|
> |arithmetic|12|6|
> |inequality|5|3|
> |geometry|1|6|
> |sets/sequences|3|4|
> |trig. equations|2|3|
> |functions|3|2|
> |combinatorics|0|9|
> |others|3|1|
>
> Table B: Topic Distribution of Unverified Steps
> |Topic|MATH|AIME|
> |---|---|---|
> |geometry|16|14|
> |trig.|6|5|
> |functions|7 |3|
> |number theory|0|8|
> |sets/sequences|4|9|
> |equations/polynomials|7|6 |
> |others|10|5|
>
> ## W5
> Written on the verge of the DeepSeek-V3.2 release, we re-ran the main performance experiments to prolong the relevance of the work. Supporting ablation conclusions remain unchanged as they measure relative hyperparameter sensitivity, which holds valid for a tested base model. We will fix the Fig.5(b) label typo to DS-V3.2 in the revision.

---

> > ### Author Rebuttal · Reviewer_1NYm · 2026-03-31
> >
> > I have no further questions and will keep my positive score.

---

> > > ### Author Response · Authors · 2026-04-05
> > >
> > > We sincerely thank Reviewer 1NYm for the constructive feedback on our work. We will incorporate all newly discussed experiments and ablations into the revised manuscript.

---

### Official Review · Reviewer_PdeA · 2026-03-17

**Soundness:** 4
**Presentation:** 3
**Significance:** 4
**Originality:** 3
**Overall Recommendation:** 5
**Confidence:** 4

**Summary:**

Hermes is an inference-time framework for mathematical reasoning that interleaves natural-language CoT with intermediate Lean4-based formal verification. Given an input math problem, during the generation of natural-language reasoning steps, “critical” steps are autoformalized into Lean, attempted to prove (together with negation) by a prover agent, and the resulting verification signal is fed back to guide subsequent reasoning. Hermes also maintains a retrieval-based memory of previously validated steps to improve cross-step consistency. The paper evaluates this framework on MATH500, AIME’25, CollegeMath, and HARDMath2 using several base LLMs and compares against zero-shot CoT, majority vote, reward-model-based Best-of-N selection, and prior baselines incorporating formal reasoning such as Safe. The paper claims that Hermes improves final-answer accuracy while using substantially fewer tokens and lower estimated inference cost than reward-based multi-sample methods, with large gains on harder benchmarks.

**Compliance With Llm Reviewing Policy:**

Affirmed.

**Final Justification:**

Rebuttal addressed my concerns and questions, but did not change my assessment overall. I am keeping my original positive score.

**Key Questions For Authors:**

1. On average, how many verifier calls are made per problem, and what fraction of the overall reasoning chain is formally checked? Reporting this would help clarify how much of Hermes’s performance gain comes from the Lean 4 module.
2. I am unsure about the comparability assumption behind Theorem 4.1. Why is it reasonable to use the same output length `N` across Hermes and ORM/PRM methods, given that Hermes introduces verifier interactions, retries, and context growth that are likely very different from ordinary sampled CoT traces? I would find direct empirical efficiency statistics more convincing here than the current theoretical comparison.
3. Theorem 4.1 suggests that Hermes should fall between ORM and PRM in token complexity, yet Figure 3 groups ORM/PRM/Safe into a single @5 cost block. Could the authors clarify which methods are included there and how that empirical grouping should be interpreted relative to the theorem’s claimed ordering? This would help convince the reader of the assumptions made in Theorem 4.1, given the empirical evidence.

**Limitations:**

yes

**Strengths And Weaknesses:**

# Strengths

**Novel and Modular System Design:** The core method design of interleaving informal reasoning with intermediate formal verification is compelling, and the prove-or-counterprove component allows for maximising feedback given back into the reasoning loop. The method is modular and swappable across different formalizers and provers, which makes the framework extensible beyond the exact components used in this paper.

**Evaluation Breadth and Analysis:** The paper considers multiple benchmarks and LLM backbones and shows greatly improved performance rather than demonstrating the method in only one narrow setting. It also compares to and beats both generic reasoning baselines and Lean-based baselines such as Safe and Safe*. In addition, the ablations in Table 2 and the `Kf/Kp` sensitivity study in Table 3 are particularly useful and provide great insight in how to best allocate compute in the Lean4-integrated module.

**Useful Qualitative Analysis:** The appendix example of Hermes correcting a concrete reasoning error is greatly effective and helps illustrate the intended benefit of the method in practice. The additional breakdown by math topic is also useful, since it gives further insight into where the method seems to help most and supports the paper’s practical claims beyond a single aggregate accuracy number.

**Great Overall Presentation, Writing Clarity and Figures:** The paper is clearly written and generally easy to follow. The method is presented with a clean modular decomposition, and Figures 1 and 2 do a good job of making the overall pipeline and proof-checking loop understandable. The notation and section organisation are also strong overall, which is helpful for a paper with multiple moving parts in terms of design and evaluation.

# Weaknesses

**Critical-Step Selection and Verification Coverage** A weakness of the paper is that the base model appears to self-select when to invoke the verifier on “critical” steps, but the paper does not state and discuss this clearly enough. This leaves important questions unanswered: what counts as a critical step in practice, how often the tool is called, what fraction of the reasoning chain is actually verified, and how sensitive performance is to this invocation behaviour. A more robust treatment of tool-use frequency and coverage would make the method much easier to evaluate.

**Fallback Behaviour:** Related to the above, the paper fails to discuss cases where the Lean module fails, and the system continues with self-verification or cautious reasoning. Further analysis of such cases would greatly improve the paper and would answer how often Hermes is genuinely benefiting from Lean-based supervision versus effectively reverting to ordinary LLM reasoning with occasional tool use.

**Minor Presentation and Editing Issues:** Although the paper is generally polished, there are a number of visible wording or formatting issues (e.g. `higherKf`, `complxity`, `emoving`, and repeated `asympytotic` in the appendix tables).

---

> ### Author Rebuttal · Authors · 2026-03-31
>
> We thank the reviewer for the constructive feedback. Please find our responses to the comments. All new results and clarifications will be added in the revision.
>
> ## W1, Q1
> Hermes does not use a separate hand-crafted detector for critical steps. Rather, the reasoning model decides during inference whether to invoke tool usage, consistent with standard agentic works that treat tool use as a sequential decision-making process [1] within the reasoning trace. Concretely, Hermes instantiates this via a template-driven tool-usage paradigm [2], in which the model interacts with tools via a Thought–Action–Observation format.
>
> [1] https://arxiv.org/pdf/2412.04141
>
> [2] https://arxiv.org/pdf/2402.16696
>
> Moreover, we conducted an analysis of the frequency of tool invocation. Hermes, on average, **invokes the verifier 5.5 and 10.4 times on MATH and AIME**, respectively. It suggests that on more challenging benchmarks, the model relies more on Hermes and divides the problem into finer-grained sub-steps.
>
> To better understand what counts as a “critical” step in practice, we manually classified the topics of the formalized steps, shown in Table A. It shows that the LLM uses Lean primarily to verify its own solutions to the presented equations, predominantly polynomials. Lean is often invoked when the model faces a long string of arithmetic computations, while other steps are verified less frequently. We hope this provides more insight into what the LLM perceives as a “worthy” math step.
>
> Table A: Topic Distribution of Formalized Steps
> |Topic|MATH|AIME|
> |---|:---:|:---:|
> |equations/polynomials|21|16|
> |arithmetic|12|6|
> |inequality|5|3|
> |geometric identites|1|6|
> |sets/sequences|3|4|
> |trigonometric equations|2|3|
> |functions|3|2|
> |combinatorics|0|9|
> |others|3|1|
>
> To quantify the portion of the informal reasoning trace that is verified by Hermes, we report Average Verification Coverage. Here, a step is regarded as verified if it is successfully formalized, proved, and validated in Lean, yielding either a formal proof or a counter-proof. According to our evaluation, **Hermes covers 87% and 76% on MATH and AIME**, respectively, providing a formal grounding signal that reduces reasoning drift.
>
> ## W2
> To provide further insight into failure cases, we sampled math steps for which Hermes returned verification failures and classified them in Table B. We observe that geometry is one of the most challenging domains for Lean to verify, which aligns with the current state of Mathlib. **_Please also kindly refer to our response to 1NYm [W4] for an extended analysis_** of these cases. As Lean and Mathlib are constantly evolving, such failure cases will naturally decrease due to library saturation.
>
> Table B: Topic Distribution of unproven math steps
> |Topic|MATH|AIME|
> |---|---|---|
> |geometry|16|14|
> |trigonometry|6|5|
> |functions|7|3|
> |number theory|0|8|
> |sets/sequences|4|9|
> |equations/polynomials|7|6|
> |others|10|5|
>
> ## W3
> We thank the reviewer for spotting presentation mistakes. We will fix them in the revised manuscript.
>
> ## Q2
> We thank the reviewer for the comment regarding the comparability of these methods in Thm 4.1. In our analysis, we define $N$ as the total upper limit on completion tokens, the token ceiling, for the entire reasoning process, which accounts for interactions and retries. Since quantifying an exact $N$ for each individual model is difficult and a mean-based analysis is less rigorous, we modeled the worst-case scenario by setting $N$ to a maximum value across all modules. While models rarely generate up to this absolute limit in practice (particularly the formalization and prover modules), this approach establishes a formal theoretical ordering of the methods' complexities. We complement this theoretical bound with the direct empirical efficiency statistics in Tables 1 and 4, and Figs. 3, 4, and 5 in the main text.
>
> ## Q3
> Thanks for the observation. Thm 4.1 and Fig. 3 measure two complementary aspects of cost. Our theoretical analysis in Thm 4.1 evaluates the overall sampling costs across the whole inference system, where Hermes does indeed fall in between ORM and PRM.
>
> In contrast, Fig. 3 specifically reports only the reasoning LLM’s inference token cost. We grouped ORM, PRM, and Safe into a single @5 cost block in this figure because these current reward-based approaches all require sampling from the base reasoning model multiple times. Our motivation for isolating this metric is practical: reasoning models are usually the heaviest, most expensive component of the system, and are very often accessed via APIs. We wanted to highlight how generating multiple full traces ramps up this specific base cost. We will make the distinction between the overall theoretical ordering and this empirical reasoning-token grouping clearer in the revision.

---

> > ### Author Rebuttal · Reviewer_PdeA · 2026-04-03
> >
> > I thank the authors for their follow-up work on providing new evaluations and analysis, strengthening the contribution of the paper. Given the response, the authors seem to agree that the theoretical grounding of the token-complexity analysis is weak "models rarely generate up the absolute limit in practice". I would rather see this be moved to the appendix or removed from the paper entirely, as it is distracting and not additive in my assessment.
> >
> > Other than that, I have no further questions and will keep my initial positive score.

---

> > > ### Author Response · Authors · 2026-04-05
> > >
> > > We sincerely thank Reviewer PdeA for the constructive review and the feedback on our response. We will make revisions to the text as discussed in our responses to the comments and questions.

---

### Official Review · Reviewer_zAqK · 2026-03-17

**Soundness:** 2
**Presentation:** 2
**Significance:** 3
**Originality:** 3
**Overall Recommendation:** 3
**Confidence:** 3

**Summary:**

The paper introduces Hermes: a framework for solving reasoning-intensive problems by interleaving informal and formal reasoning and getting correctness feedback from Lean. Concretely, when solving a problem requires N reasoning steps, some of these steps are autoformalized and formally verified with Lean. Based on the feedback, the base model may proceed further, revise a faulty reasoning step, or self-verify a formally unverified step.

**Compliance With Llm Reviewing Policy:**

Affirmed.

**Key Questions For Authors:**

1. How do you isolate reasoning steps to verify?
2. Could you provide 5 examples of reasoning steps which you pass to autoformalizer?
3. On average, how many reasoning steps do you verify with Hermes for one reasoning trace from the base model?
4. Could you provide statistics how often reasoning steps are formally validated, invalidated, left undecided?
5. Could you provide one example of full run of Hermes for each of the four benchmarks?
6. Could you contrast your method with Safe?

**Limitations:**

yes

**Strengths And Weaknesses:**

**Strengths**

The topic of combining formal and informal reasoning ifs definitely an interesting one.

The authors provide quite extensive experiments and ablations.

The design of the framework is quite novel.

**Weaknesses**
1. Some important details of the method are not described: see questions.
2. It seems that the generation FLOP budged in experiments in Table 1 is not aligned. For instance, CoT@1 requires generating one sample. Hermes@1 requires one sample from the base model, but additionally N * 4 samples from the autoformalizer and N * 4 samples from the prover, where N is the number of reasoning steps to verify.
3. It would be good to contrast Hermes with Safe: another work combining informal reasoning with Lean verifier.

**Minor**
The font in Fig. 3 is too small.
previously context --> previousl context

---

> ### Author Rebuttal · Authors · 2026-03-31
>
> We thank the reviewer for the constructive feedback. Please find our responses to the comments below. All new results and clarifications will be added in the revision.
>
> ## W2
> Table 1 reports the final accuracy of each method, while Fig.5 compares methods' underlying FLOPs budgets. Hermes@1 indeed involves additional generations from the formalizer and prover beyond a single base-model sample. However, sample count does not translate directly into FLOP cost. As described in Sec. 5.4, FLOPs also depend on parameter count, model size, and token count. In addition, the sampling parameters $N, K_p, K_f$ specify only the upper bound, rather than the number used uniformly at every step.
>
> ## W3, Q6
> Safe ranks multiple CoTs using retrospective step-by-step verification and reward modeling. Conversely, Hermes uses Lean for real-time steering of a single CoT, bypassing the need for sampling multiple CoTs. Benchmarks in Tables 1 and 4 show that Hermes achieves similar or better accuracy than both Safe and our controlled baseline (Safe*), validating its effectiveness and efficiency.
>
> ## Q1
> Hermes does not use a separate hand-crafted detector to isolate steps for verification. Instead, the reasoning model decides during inference whether to invoke tool usage, leveraging tool use as a standard capability of agentic LLMs and consistent with standard agentic works that treat tool use as a sequential decision-making process [1] within the reasoning trace. Concretely, Hermes instantiates this via a template-driven tool-usage paradigm [2], in which the model interacts with tools via a Thought-Action-Observation format until reaching the final answer.
>
> [1] https://arxiv.org/pdf/2412.04141
>
> [2] https://arxiv.org/pdf/2402.16696
>
> To better understand the types of steps isolated for formal verification, we manually classified the topics of the mathematical steps that trigger verification, as shown in **_Tables A and B of our response to 1NYm_**. The data shows that the LLM uses Lean primarily to verify its solutions to equations, predominantly polynomials. Moreover, Lean is often invoked when the model faces long strings of arithmetic computations, while other steps are verified less frequently. Since the boundary between critical and non-critical steps is subjective, we do not impose a fixed external rule; instead, we report the model’s empirical behavior in practice.
>
> ## Q2
> We will include intermediate math step examples in the revised manuscript.
>
> ## Q3
> As discussed in our response to Q1, Hermes does not verify every reasoning step. Instead, the reasoning model decides during inference when to invoke formal verification.
>
> To provide objective insight, we conducted an empirical analysis of tool call frequency. Hermes, on average, invokes the verifier 5.5 and 10.4 times on MATH and AIME, respectively. It suggests that on more challenging benchmarks, the model relies more heavily on Hermes and divides the problem into finer-grained sub-steps.
>
> To quantify the proportion of the informal reasoning trace that is verified by Hermes, we report Average Verification Coverage in Table A. In our framework, a step is regarded as verified if it is successfully formalized, proved, and validated in Lean, yielding either a formal proof or a counter-proof. For such steps, the correctness signal is grounded in Lean’s trusted kernel.
>
> Table A: Average Verification Coverage
> ||MATH|AIME|
> |---|---|---|
> |DS-V3.2|87%|76%|
>
> As shown, Hermes successfully verifies a sizable majority of the reasoning steps. This high coverage provides a formal grounding signal that empirically reduces the likelihood of reasoning drift. Even without a single full formal proof for the entire trace, the informal reasoning remains anchored to formally verified subgoals. The effectiveness of this local formal grounding is reflected in the improved final-answer accuracy reported in Table 1. We will include this analysis in the revised paper to further clarify the reliability of Hermes.
>
> ## Q4
> Thanks for the suggestion. We report the distribution of validated/invalidated/unverified proofs on MATH and AIME in Table B, and will add this to the revised paper.
>
> Table B: Ratio of Hermes outputs across the whole dataset
> |Dataset|Validated|Invalidated|Unverified|
> |---|---|---|---|
> |MATH|71%|4%|25%|
> |AIME|66%|4%|30%|
>
> These global statistics are aggregated over randomly sampled agent outputs, rather than averaged per problem. We observe that in the vast majority of mathematical steps, Hermes can obtain a valid Lean proof. Moreover, the proportion of invalidated steps, where Hermes finds a counter-example or counter-proof, is small. This suggests that the reasoning LLM generally constructs effective intermediate steps, while Hermes helps to identify the subset of flawed steps that could otherwise lead to reasoning drift or hallucinated outputs.
>
> ## Q5
> Thanks for the suggestions. Due to the rebuttal length limit, we will include additional sample runs in the revised paper.

---

> > ### Author Rebuttal · Reviewer_zAqK · 2026-04-04
> >
> > I'm partially satisfied about the answer however still not convinced enough. It's difficult to judge from the results how Hermes improves the results given similar inference-time compute budgets.
> >
> > For instance, it may be the case that if we align budgets of majority voting and Hermes, the former may be better.
> >
> > Also, what's the FLOP cost difference between Hermes and Hermes+Skywork?
> >
> > And seeing some examples of reasoning steps being formalized would be crucial for me to better assess the work.

---

> > > ### Author Response · Authors · 2026-04-05
> > >
> > > We thank the reviewer for the follow up questions regarding our work.
> > >
> > > > **For instance, it may be the case that if we align budgets of majority voting and Hermes, the former may be better.**
> > >
> > > Per the reviewer’s request, we report Hermes@5+Majority to demonstrate the scaling behavior when multiple agentic traces are aggregated without external reward model.
> > >
> > > Table X: Accuracy Comparison under Aligned Sampling Budgets (@5)
> > > | Method | Qwen3-8B (MATH) | Qwen3-8B (AIME) | Qwen3-8B (CM) | Qwen3-8B (HM2) | o3-mini (MATH) | o3-mini (AIME) | o3-mini (CM) | o3-mini (HM2) | DS-V3.2 (MATH) | DS-V3.2 (AIME) | DS-V3.2 (CM) | DS-V3.2 (HM2) |
> > > |---|---|---|---|---|---|---|---|---|---|---|---|---|
> > > | **CoT@5+Majority** | 87.0 | 16.7 | 70.3 | 4.7 | 96.8 | 70.0 | 76.0 | 22.7 | 97.6 | 60.0 | 80.8 | 26.1 |
> > > | **HERMES@5+Majority** | 93.0 | 33.3 | 75.8 | 7.6 | 97.0 | 86.7 | 79.4 | 31.8 | 98.6 | 76.7 | 85.2 | 35.1 |
> > >
> > > We observe a pattern consistent with standard CoT: increasing the sampling budget and incorporating a selection rule (such as Majority Voting or Best-of-N with Skywork) significantly improves performance. Crucially, because the token budget of Hermes@5 is comparable to that of CoT@5 (according to Figures 3 and 5), this comparison demonstrates that the advantages of our method persist regardless of the specific selection rule or compute ceiling. We will include these results in the revised manuscript.
> > >
> > > > **Also, what's the FLOP cost difference between Hermes and Hermes+Skywork?**
> > >
> > > Figure 5 reports the compute budget for Hermes@5. Since Skywork is an Outcome Reward Model (ORM) that produces only a single scalar score, its FLOP contribution to the overall budget is extremely small. Below, we calculate the compute cost (consistent with Figure 5) based on Skywork’s configuration, assuming a maximum context length of 8,192 tokens:
> > >
> > > $$C_{forward} = 2N + 2n_{layer}n_{ctx}d_{attn} = 2 * 8 * 10^9 + 2 * 32 * 8192 * 4096 \approx 0.02 \times 10^{12} \approx 0.02 \text{ TFLOPs}$$
> > >
> > > Since we run Skywork five times, the overall energy consumption is approximately 0.1 TFLOPs for Hermes@5.
> > >
> > > > **seeing some examples of reasoning steps being formalized would be crucial for me to better assess the work.**
> > >
> > > As per the reviewer’s request we report six math step examples sampled from MATH and AIME below:
> > >
> > > Table Y: Informal and formal math steps from MATH and AIME generated by DS-V3.2
> > >
> > > | Dataset | Informal step | Formal step |
> > > |---|---|---|
> > > | MATH500 | Given that: - f(f(1)) = 4. - f(1) = 2 when f(x) = 2^x.  Prove that: f(f(f(1))) = f(4) = 2^4 = 16 | ```theorem test (f : ℕ → ℕ) (h1 : f (f 1) = 4) (h2 : ∀ x, f x = 2^x) : f (f (f 1)) = 16 ∧ f 4 = 16 := by sorry``` |
> > > | MATH500 | Given 5 shirts, 6 pairs of pants, and 8 hats, the total number of outfits (one of each) is 5 × 6 × 8 = 240. | ```theorem test : let shirts := Finset.range 5 let pants := Finset.range 6 let hats := Finset.range 8 (shirts ×ˢ pants ×ˢ hats).card = 240 := by sorry``` |
> > > | MATH500 |  Given that 24^(24^24) is even, and the pattern shows that 24ⁿ ≡ 6 mod 10 when n is even, therefore 24^(24^(24^24)) ≡ 6 mod 10. | ```theorem test : (Even (24^(24^24))) → (∀ n : ℕ, Even n → 24^n ≡ 6 [MOD 10]) → (24^(24^(24^24)) ≡ 6 [MOD 10]) := by sorry``` |
> > > | AIME | Given a positive integer N = 3^4 × 5^2, the set A of positive integer divisors of N consists of all numbers of the form 3^a × 5^b where 0 ≤ a ≤ 4 and 0 ≤ b ≤ 2. The number of such divisors is (4+1) × (2+1) = 15. | ```theorem test : let N := 3^4 * 5^2 let A := {d : ℕ \| ∃ a b, a ≤ 4 ∧ b ≤ 2 ∧ d = 3^a * 5^b} (∀ x, x ∈ Nat.divisors N ↔ x ∈ A) ∧ ((Nat.divisors N).card = (4 + 1) * (2 + 1)) ∧ ((4 + 1) * (2 + 1) = 15) := by sorry``` |
> > > | AIME | Given that: - If d divides 3(d+1)(d^2-4d+13) and d is coprime to d+1, then d divides 3(d^2-4d+13).  Prove that: The positive integers n such that n+2 divides 3(n+3)(n^2+9) are n = 1, 11, and 37. Their sum is 1 + 11 + 37 = 49. | ```theorem test : (∀ d : ℤ, d ∣ 3 * (d + 1) * (d^2 - 4*d + 13) → Int.gcd d (d + 1) = 1 → d ∣ 3 * (d^2 - 4*d + 13)) → {n : ℕ \| n > 0 ∧ (n + 2 : ℤ) ∣ 3 * (n + 3) * (n^2 + 9)} = {1, 11, 37} ∧ 1 + 11 + 37 = 49 := by sorry``` |
> > > | AIME | Prove that: Given that the total number of ways is 9! × 56 × (3!)^6, we compute the prime factorization: 9! = 2^7 * 3^4 * 5 * 7; 56 = 2^3 * 7; (3!)^6 = 2^6 * 3^6. Multiplying yields 2^(7+3+6) * 3^(4+6) * 5^1 * 7^(1+1) = 2^16 * 3^10 * 5^1 * 7^2. | ```theorem test : 9! = 2^7 * 3^4 * 5 * 7 ∧ 56 = 2^3 * 7 ∧ (3!)^6 = 2^6 * 3^6 ∧ 9! * 56 * (3!)^6 = 2^16 * 3^10 * 5 * 7^2 := by sorry``` |
> > >
> > > We will include these math step examples in the revised manuscript.

---

### Official Review · Reviewer_2tBv · 2026-03-22

**Soundness:** 3
**Presentation:** 3
**Significance:** 1
**Originality:** 1
**Overall Recommendation:** 2
**Confidence:** 5

**Summary:**

This paper presents the Hermes framework, which integrates reasoning LLMs with formalization module, theorem proving module, feedback module, and memory blocks, enabling an interesting mix of informal reasoning and formal reasoning in solving math problems. All modules are specialized for the Lean environment. Formalization module and theorem proving modules can be instantiated with existing works on auto-formalization and theorem proving in Lean. Experimental evaluations show that Hermes outperforms existing reward model-based approaches, and furthermore, Hermes combined with the Skywork reward model achieves the strongest result.

**Compliance With Llm Reviewing Policy:**

Affirmed.

**Final Justification:**

After reading the rebuttal and re-reading the original paper, I still feel this work is over claiming many things. Many prior works fall into the tool-assisted agent interleaving informal reasoning and formal theorem proving, this work is certainly not the first. Claiming so in the abstract is misleading. Clarification basically argues that differences can be made by further inspecting 1) whether it is Lean-based or Isabelle-based (like DSP), 2) the "direction" of interleaving (e.g., informal to formal or formal to informal), 3) whether feedback might be considered as "real-time" or not. Taking all these factor into consideration would make this work sound unique, which by itself is a weak argument. Similar over claim is made in the end of intro when highlighting contributions, e.g., "Hermes improves average accuracy by 23% across four benchmarks", which is in fact the _average_ of _relative percentage improvement_. A metric like this could easily amplify outliers, for instance, CoT@1 with Qwen3-8B achieves 4.3% on HM2, while Hermes@1 with Qwen3-8B achieves 6.6%, Hermes@5+Skywork with Qwen3-8B achieves, in other words, the latter two improve (relative) accuracy by 53% and 255%, respectively. Given all such observations, the claimed contributions and presented results remain quite fishy to me.

**Key Questions For Authors:**

The problem formulation (section 3) suggests that the trace consists of T steps (not tokens). How to detect those reasoning steps (from tokens)? How to decide whether a reasoning step is critical (thus worth formalizing) or not (thus reasoning informally)?

What concrete assumptions (e.g., tool use capability) about the reasoning LLMs?

How reliable is the back-translation in terms of properly aligning formalized theorem with the original informal one in natural language?

Are there any formal guarantees regarding the correctness of the entire reasoning traces?

**Limitations:**

yes

**Strengths And Weaknesses:**

Strengths
- this paper concerns an important challenge, specifically, how to effectively combine informal reasoning with formal reasoning
- the Hermes framework is carefully illustrated with nice visualizations for each module
- the experimental evaluations consider multiple aspects like accuracy, token usage, FLOPs, scaling behavior, and ablation studies

Weaknesses
- the general idea of mixing informal reasoning and formal reasoning is not new and has been explored in several prior works (e.g., DSP, Lean-STaR); claiming this work as the first agent that interleaves informal reasoning with formally verified proofs in lean is over-selling and problematic
- both the abstract and conclusion give the impression that hallucination is eliminated through formal theorem proving in Lean, which is however not the case, since only a fraction of intermediate steps are formalized, and the formalization correctness relies on LLM-as-a-judge; as a result, even if the final outcome matches the ground truth, there are no correctness guarantee regarding the reasoning process
- the essential improvement is not as promising as highlighted in the introduction, but instead, they are quite incremental; concretely, reporting the average of relative percentage is meaningless or misleading. For instance, the increase of 6.6% to 10% would contribute to more than 50% in terms of relative percentage improvement, greatly biasing the average. Calling absolute 1% improvement (say from 8% to 9%) a dramatic 12.5% (relative) improvement isn't technically wrong but misleading.
- given the modest improvement, the overall design seems to be too complicated, especially Hermes is further integrated with reward models

---

> ### Author Rebuttal · Authors · 2026-03-31
>
> We thank the reviewer for the constructive feedback. Please find our responses to the comments. All new results and clarifications will be added in the revision.
>
> ## W1
> DSP uses informal proofs to guide formal sketches, while Lean-STaR interleaves informal thoughts step-by-step before each formal step. In both works, informal reasoning serves to facilitate formal proof construction.
>
> In contrast, Hermes uses formal verification to verify and steer informal reasoning. During inference, it converts candidate steps into Lean statements and verifies them. Then it feeds the result back to guide the next step. This verification-in-the-loop helps to prevent the model from diverging into hallucinated natural language reasoning paths.
>
> ## W2, Q3, Q4
> Thanks for insightful questions. Hermes aims to reduce hallucinations and reasoning drift during the informal reasoning process through formal grounding, rather than eliminating them.
>
> To support reliable formalization, we follow common practice in prior works like HERALD. **We conducted a human double-review by Lean experts on steps sampled from MATH and AIME, obtaining alignment accuracies of 86% and 80%**. It indicates that our formalization is reasonably robust.
>
> Since Hermes operates in an informal mathematical space rather than producing an end-to-end formal proof, it does not provide formal guarantees for the entire informal reasoning trace. However, it provides a local formal guarantee for those intermediate steps that are successfully verified in Lean, which helps to reduce reasoning drift.
>
> To quantify the portion of the informal reasoning trace that is verified by Hermes, we report Average Verification Coverage. Here, a step is regarded as verified if it is successfully formalized, proved, and validated in Lean, yielding either a formal proof or a counter-proof. According to our evaluation, **Hermes covers 87% and 76% on MATH and AIME**, providing formal grounding signals.
>
> ## W3
> We appreciate the suggestion. We will revise the manuscript to prioritize absolute percentage improvements throughout for better clarity and transparency. We also respectfully clarify that the improvements are not merely marginal; please kindly refer to our response to [W4] below for clarification of the experimental setting.
>
> Under this setting, **Hermes delivers gains across different models and datasets, and remains consistently stronger than the baselines** overall, showing the effectiveness of our method. For instance, on AIME, Hermes@1 improves DeepSeek-V3.2 (CoT@1) from 50% to 70%, a notable 20pt absolute gain. On HM2, Hermes@1 improves Qwen3-8B from 4.3% to 6.6%, a 2.3-point absolute gain, which is still meaningful given the difficulty of the benchmark and the model scale.
>
> ## W4
> Table 1 compares Hermes and 8 baselines (also see Sec. 5.1). Specifically, our primary comparison is the standalone Hermes@1 setting against the base model (CoT@1), prior Lean-based Safe, and ORM/PRM-based methods. **Averaged over all model-dataset settings, Hermes@1 improves over CoT@1 by 7.8 absolute points**.
>
> The additional Hermes@5+Skywork that combines Hermes with reward models is an optional extension, included to show that Hermes is modular and complementary to existing SOTA inference methods.
>
> ## Q1
> Thanks for the question. Hermes does not use a separate hand-crafted detector for critical steps. Rather, the reasoning model decides during inference whether to invoke tool usage, consistent with standard agentic works that treat tool use as a sequential decision-making process [1] within the reasoning trace. Concretely, Hermes instantiates this via a template-driven tool-usage paradigm [2], in which the model interacts with tools via a Thought–Action–Observation format until reaching the final answer.
>
> [1] https://arxiv.org/pdf/2412.04141
>
> [2] https://arxiv.org/pdf/2402.16696
>
> Moreover, we conducted an analysis of tool invocation frequency. Hermes, on average, invokes the verifier 5.5 and 10.4 times on MATH and AIME, respectively. It suggests that on more challenging benchmarks, the model relies more heavily on Hermes and divides the problem into finer-grained sub-steps.
>
> Regarding the second part of Q1, we manually classified the topics of the mathematical steps (**_reported in Table A of our response to PdeA_**). It provides a clearer picture of what the reasoning LLM perceives as a mathematical step worth formalizing. It shows that the LLM uses Lean primarily to verify its own solutions to the presented equations and predominantly polynomials. Moreover, Lean is often invoked when the model faces a long string of arithmetic computations, while other steps are verified less often. Since the boundary between critical and non-critical steps is subjective, we do not impose a fixed external rule; instead, we report the model’s empirical behavior.
>
> ## Q2
> The assumption we make about the reasoning LLMs is the ability to invoke tools, which is now a standard capability of agentic LLMs.

---

> > ### Author Rebuttal · Reviewer_2tBv · 2026-04-03
> >
> > I largely remain confused thus unconvinced about various metrics the authors try to highlight, including initially average of relative percentage improvements, and newly suggested average of model-dataset setting (between different setups, Lean-based vs non-lean). A fair comparison shall be in the same setup (e.g., Lean-based method),  Hermes@1 vs "CoT@1+safe/safe*" instead. Whether using average of relative percentage improvements or another novel illustration metric does not affect the actual results themselves shown in Table 1. From Table 1, we can see that the improvement of Hermes over the baseline CoT@K+safe is not that significant (except for two model-dataset settings, i.e., Qwen3-8B/AIME and DeepSeek-V3.2/AIME), whereas enabling Skywork (ORM/PRM approach) dramatically boost the performance. The main technical novelty of this work is some form of synergistic combination of informal reasoning and formal reasoning. This combination has been explored and the newly proposed combination does not significantly improve existing combinations.   Certain novel metrics may amplify the effects of 1-2 outliers and make the marginal improvement appear to be way more significant, which however does not impress the reviewer in that way.

---

> > > ### Author Response · Authors · 2026-04-05
> > >
> > > Regarding the reviewer’s feedback on our response, we would like to provide the following clarifications.
> > >
> > > # On Evaluation Metrics and Performance Comparison
> > > **We respectfully note that our presented evaluation metric and settings align with standard evaluation practice for assessing performance in the literature.** We compared our method against 8 baselines on 4 datasets using 3 models of different scales, and our presentation of results is consistent with prior works.
> > >
> > > We kindly refer the reviewer to the references for our baselines, including Skywork (Liu et al., 2024), ArmoRM (Wang et al., 2024a), Shepherd (Wang et al., 2024b), RLHFlow (Dong et al., 2024), and Safe (Liu et al., 2025) to verify our standard evaluation protocol. _Given that our setting is consistent with the established practices in this field, we believe the reviewer’s concerns would equally apply to prior publications that adopted the same setting._
> > >
> > > # On the novelty of using formal verification to guide informal reasoning during inference in real time
> > >
> > > We respectfully disagree with the reviewer’s assessment of our work’s novelty.
> > >
> > > Prior works on informal-formal synergy spans two different tasks (as described in Secs. 1-2 and summarised in Table X below):
> > > - (a) informal mathematical reasoning, whose goal is to produce a natural-language reasoning trace leading to the correct final answer, and
> > > - (b) automated theorem proving, where the goal is to generate a formal Lean proof.
> > >
> > > **Hermes belongs to (a) informal mathematical reasoning.** A majority of the prior methods discussed here instead belong to (b) automated theorem proving. For example, DSP/DSP+, Lean-STaR, APOLLO, REAL-Prover, and DeltaProver use informal reasoning to support formal proof construction, not to improve a natural-language reasoning trace for final-answer computation. Hermes targets a different task than Lean-STaR or DSP; its goals and design target the hallucinations and semantic drift present in informal CoTs, with the goal of improving the correctness of the final answer.
> > >
> > > **Within (a) informal mathematical reasoning, Hermes differs fundamentally from prior methods: Hermes is the first method to use Lean-based verification to directly steer informal reasoning during inference via real-time feedback.** Existing informal math methods typically rely on post-inference selection (e.g., Best-of-N) over multiple sampled Chain-of-Thought (CoT) traces. Among them, only Safe uses Lean-based formal verification. Specifically, Safe reranks multiple CoTs via retrospective reward modeling trained on step-level Lean verifications, whereas ORM/PRM-based methods such as Skywork, ArmoRM, Shepherd, and RLHFlow do not use formal verification; they are reward models that score multiple completed CoTs and select the highest-scoring one. The novelty of Hermes lies not merely in combining informal and formal reasoning, but in introducing a novel inference-time paradigm in which interleaved formal verification guides a single ongoing reasoning trace.
> > >
> > > Table X: Comparison of prior work on informal-formal synergy. “Task” indicates whether the method targets informal mathematical reasoning or automated theorem proving; “Primary mechanism” indicates the main type of method proposed. IMR stands for Informal Math Reasoning and FTP is for Formal Theorem Proving.
> > >
> > > | Paper | Task | Primary Mechanism | Short Description |
> > > |---|---|---|---|
> > > | HERMES (ours) | IMR | During-Inference Verification | Real-time tool-based Lean verification during informal inference. |
> > > | Safe (Liu et al., 2025) | IMR | Post-Inference Selection + Training | Reward model ranking CoTs via formalized Lean training data. |
> > > | DTV (Peng et al., 2024) | IMR | Post-Inference Selection | Ranks CoTs post-inference using high-level Isabelle proofs. |
> > > | APOLLO (Ospanov et al., 2025) | FTP | Recursive Inference | Recursive sub-goal decomposition via compiler feedback. |
> > > | Lean-STaR (Lin et al., 2025) | FTP | Training | Fine-tunes models via expert iteration to interleave informal and formal steps. |
> > > | REAL-Prover (Shen et al., 2025) | FTP | Training | Retrieval-augmented approach appending Mathlib context. |
> > > | Draft, Sketch, and Prove (DSP/DSP+) (Jiang et al., 2023) (Cao et al., 2025) | FTP | Inference | Maps informal drafts to formal sketches for proof construction. |
> > > | Delta Prover (Zhou et al., 2025) | FTP | Iterative-Inference | Reflective decomposition and iterative proof repair in Lean 4. |
> > >
> > > # On “CoT@1+Safe” experiment
> > >
> > > We would like to clarify that the suggested 'CoT@1 + Safe/Safe*' is not a meaningful baseline. As explained in our response above on the method's novelty and illustrated in Table X, Safe and all other tested baselines, such as Skywork, ArmoRM, Shepherd, and RLHFlow, are post-inference ranking methods. They require multiple sampled CoTs and then select the highest-scoring CoT. **Applying them to a single sample (@1) is not meaningful and reduces them to the underlying CoT@1 baseline in Table 1 of our manuscript.**

---

### Decision · Program_Chairs · 2026-04-30

**Decision:**

Accept (regular)

**Comment:**

This paper proposes a reasonable agentic framework for interleaving informal reasoning with formal verification, with consistent improvements across multiple benchmarks. While novelty is somewhat incremental and some claims are overstated, the empirical results are solid and broadly supported. The approach is likely to be useful in practice, and I recommend acceptance.